# CD4 and CD8 co-receptors modulate functional avidity of CD1b-restricted T cells

Charlotte A. James[1], Yuexin Xu[2], Melissa S. Aguilar[3], Lichen Jing[3], Erik D. Layton[3], Martine Gilleron[4], Adriaan J. Minnaard [5], Thomas J. Scriba [6], Cheryl L. Day[7], Edus H. Warren[1,2,3,8], David M. Koelle[2,3,9,10,11] & Chetan Seshadri [3,12✉]

T cells recognize mycobacterial glycolipid (mycolipid) antigens presented by CD1b molecules, but the role of CD4 and CD8 co-receptors in mycolipid recognition is unknown. Here we show CD1b-mycolipid tetramers reveal a hierarchy in which circulating T cells expressing CD4 or CD8 co-receptor stain with a higher tetramer mean fluorescence intensity than CD4-CD8- T cells. CD4+ primary T cells transduced with mycolipid-specific T cell receptors bind CD1b-mycolipid tetramer with a higher fluorescence intensity than CD8+ primary T cells. The presence of either CD4 or CD8 also decreases the threshold for interferon-γ secretion. Co-receptor expression increases surface expression of CD3ε, suggesting a mechanism for increased tetramer binding and activation. Targeted transcriptional profiling of mycolipid-specific T cells from individuals with active tuberculosis reveals canonical markers associated with cytotoxicity among CD8+ compared to CD4+ T cells. Thus, expression of co-receptors modulates T cell receptor avidity for mycobacterial lipids, leading to in vivo functional diversity during tuberculosis disease.

[1] Molecular Medicine and Mechanisms of Disease PhD Program (M3D), Department of Pathology, University of Washington, Seattle, WA, USA. [2] Clinical Research Division, Fred Hutchinson Cancer Research Center, Seattle, WA, USA. [3] Department of Medicine, University of Washington, Seattle, WA, USA. [4] Institut de Pharmacologie et de Biologie Structurale, IPBS, Université de Toulouse, CNRS, UPS, 31077 Toulouse, France. [5] Stratingh Institute for Chemistry, University of Groningen, Groningen, The Netherlands. [6] South African Tuberculosis Vaccine Initiative and Institute of Infectious Disease and Molecular Medicine, Division of Immunology, Department of Pathology, University of Cape Town, Cape Town, South Africa. [7] Emory Vaccine Center and Department of Microbiology and Immunology, Emory University, Atlanta, GA, USA. [8] Vaccine and Infectious Disease Division, Fred Hutchinson Cancer Research Center, Seattle, WA, USA. [9] Department of Laboratory Medicine, University of Washington, Seattle, WA, USA. [10] Department of Global Health, University of Washington, Seattle, WA, USA. [11] Benaroya Research Institute, Seattle, WA, USA. [12] Tuberculosis Research and Training Center, Seattle, WA, USA. ✉email: seshadri@u.washington.edu

T cells express a T cell receptor (TCR) that mediates recognition of antigens in the context of antigen-presenting molecules. Canonically, T cells recognize peptide antigens in the context of major histocompatibility complex class I (MHC-I) or class II (MHC-II). Recognition of peptides presented by MHC-I and MHC-II is augmented by the expression of CD8 or CD4 co-receptors on the surface of the T cell, respectively. TCR co-receptors facilitate positive selection and help to define functional subsets. Co-receptors also increase TCR avidity for peptide-MHC, which has downstream effects on proliferation, memory phenotype acquisition, and functional differentiation[1–4]. Canonically, T cells that express the CD4 co-receptor can be divided into four major functional subsets: Th1, Th2, Th17, and Treg. These functional lineages are driven by the master transcription factors T-bet, GATA3, ROR-γt, and FOXP3, respectively[5–9]. Conversely, peptide-specific T cells that express the CD8 co-receptor are traditionally cytotoxic T cells, which are defined by the expression of cytolytic effector molecules, such as granzymes and perforin[10]. These classifications are not absolute as T cells can exhibit functional plasticity and alter their functional program over time[11].

A small but consistent minority of T cells that do not express either CD4 or CD8 co-receptor provided proof-of-concept for MHC-independent modes of T cell activation[12]. Specifically, invariant natural killer T (iNKT) cells recognize lipids presented by cluster of differentiation 1 (CD1) and mucosal-associated invariant T (MAIT) cells recognize metabolites presented by MHC-related protein 1 (MR1)[13,14]. On average, 15% of iNKT cells express CD4, 49% are double negative (DN), and roughly 34% express the CD8αα homodimer[15]. Among MAIT cells, 35% express CD8αα and 45% express CD8αβ[16]. There is evidence that co-receptors are actively involved in antigen recognition by iNKT and MAIT cells. CD4 potentiates iNKT cell activation leading to sustained TCR signaling and potentiation of effector responses[17]. In addition, blocking CD8 with a monoclonal antibody leads to decreased MAIT cell responses to *Escherichia coli*[18]. iNKT and MAIT cells can also be divided into distinct functional classes based on co-receptor expression. In humans, iNKT cells that express the CD4 co-receptor simultaneously secrete both Th1 and Th2 cytokines, and DN iNKT cells have a Th1 phenotype[19,20]. CD8 MAIT cells express higher levels of granulysin, granzyme B, and perforin, suggesting that they are more potently cytotoxic[21]. DN MAIT cells express less IFN-γ and more IL-17 than CD8 MAIT cells, and have a higher ROR-γt to T-bet ratio, indicative of a Th17 phenotype[21]. Notably, these functional classes exist without two pathways for selection, antigen processing, and antigen presentation as in the case of MHC-I and MHC-II. Thus, despite early reports that suggested these innate-like T cells might be limited to the minority of T cells lacking expression of co-receptors, iNKT and MAIT cells are clearly part of the majority of T cells that express either CD4 or CD8.

T cells also recognize bacterial cell wall lipids presented by Group 1 CD1 molecules (CD1a, CD1b, and CD1c), and this population of T cells may be functionally distinct from iNKT and MAIT cells[22]. The contribution of TCR co-receptors in the recognition and functional differentiation of Group 1 CD1-restricted T cells is unknown. A study of T cell clones that recognize a mycobacterial cell wall glycolipid presented by CD1b, glucose monomycolate (GMM), revealed that high-affinity T cell clones expressed the CD4 co-receptor and lower affinity T cell clones expressed either the CD8αβ heterodimer or were DN[23]. Further, T cell clones expressing high-affinity TCRs were biased towards the expression of Th1 cytokines, whereas T cell clones with lower affinity TCRs expressed Th1 and Th17 cytokines[23]. These limited data suggest that TCR co-receptors may influence the recognition of lipid antigens and the functional lineage of lipid-specific T cells.

We used T cells specific for GMM as well as sulfoglycolipids (SGL), a class of lipids that are uniquely expressed by *Mycobacterium tuberculosis* (M.tb) and presented to T cells by CD1b, as a model system to investigate the impact of co-receptor expression on T cell function[24]. We examine co-receptor expression on SGL-specific T cells directly ex vivo, as well as differences in SGL-CD1b and GMM-CD1b tetramer binding in ex vivo T cells, in vitro-derived T cell lines, and T cells transduced with an exogenous TCR. We found that SGL-specific T cells identified ex vivo by tetramer show a bias toward CD4 co-receptor expression, and that CD4 T cells bind SGL-CD1b tetramer with a higher intensity and have lower activation thresholds than T cells that express the CD8 co-receptor or are DN. Targeted transcriptional profiling studies of SGL-specific T cells revealed distinct TCR repertoires and functional programs that could be classified on the basis of CD4 and CD8 expression. These programs more closely align with functional subsets of peptide-specific T cells rather than iNKT or MAIT cells, revealing a previously unappreciated diversity among the functional profiles of CD1b-restricted T cells.

## Results

**Ex vivo co-receptor expression by SGL-specific T cells**. We first determined which co-receptor was expressed by SGL-specific T cells directly ex vivo using CD1b tetramers loaded with a biologically validated synthetic analog of SGL[25]. We examined peripheral blood mononuclear cells (PBMC) derived from five U.S. healthy donors, five South African adolescents with latent tuberculosis infection (LTBI), and five South African adolescents without LTBI (Supplementary Table 1, Supplementary Table 2, and Supplementary Fig. 1)[26]. To ensure accurate identification of antigen-specific T cells, we used a dual-tetramer labeling strategy incorporating SGL-CD1b tetramer labeled with two distinct fluorochromes and defined SGL-specific T cells as staining with both tetramers while being unlabeled by a mock-loaded CD1b tetramer (Fig. 1a, Supplementary Fig. 1). Gates defining SGL-CD1b tetramer-positive cells were set based on staining an SGL-specific T cell line and fluorescence minus one (FMO) controls (Fig. 1a)[25]. Representative staining from one individual is shown (Fig. 1b). The frequency of SGL-CD1b tetramer-positive T cells did not significantly differ between groups ($p = 0.4$, Supplementary Fig. 1).

SGL-CD1b specific T cells exhibited an 18-fold enrichment of CD4 and CD8 double-positive (DP) cells ($p = 0.003$) and a 1.5-fold enrichment of CD4 T cells ($p = 0.003$) relative to total CD3+ T cells. This was accompanied by a 5.6-fold reduction of CD8 cells ($p = 0.003$) and a 2.8-fold reduction of CD4 and CD8 double-negative (DN) cells ($p = 0.004$) (Fig. 1c). Further, SGL-CD1b specific T cells expressing any combination of co-receptors exhibit consistently higher tetramer MFI when compared to DN SGL-CD1b specific T cells ($p < 0.0001$) (Fig. 1d). We considered the possibility that the difference in tetramer MFI could be explained by differences in TCR expression. We found no differences in CD3ε expression among DP, CD4, and CD8 T cells (Fig. 1e). However, we did detect a 4% lower CD3ε MFI among DN T cells when compared to the other groups ($p = 0.05$) (Fig. 1e). For additional context, we quantified the level of CD3ε expression in all DP, CD4, CD8, and DN T cells from these donors and found that CD4 and DN T cells express higher levels of CD3 than CD8 T cells, which is consistent with published literature ($p < 0.0001$ and $p = 0.004$, respectively, Supplementary Fig. 1)[27]. Taken together, these data reveal a hierarchy in which SGL-specific T cells expressing the CD4 co-receptor are present in peripheral blood at the highest frequency and stain with SGL-CD1b tetramer at the highest MFI, followed by CD8 and DN T cells, respectively.

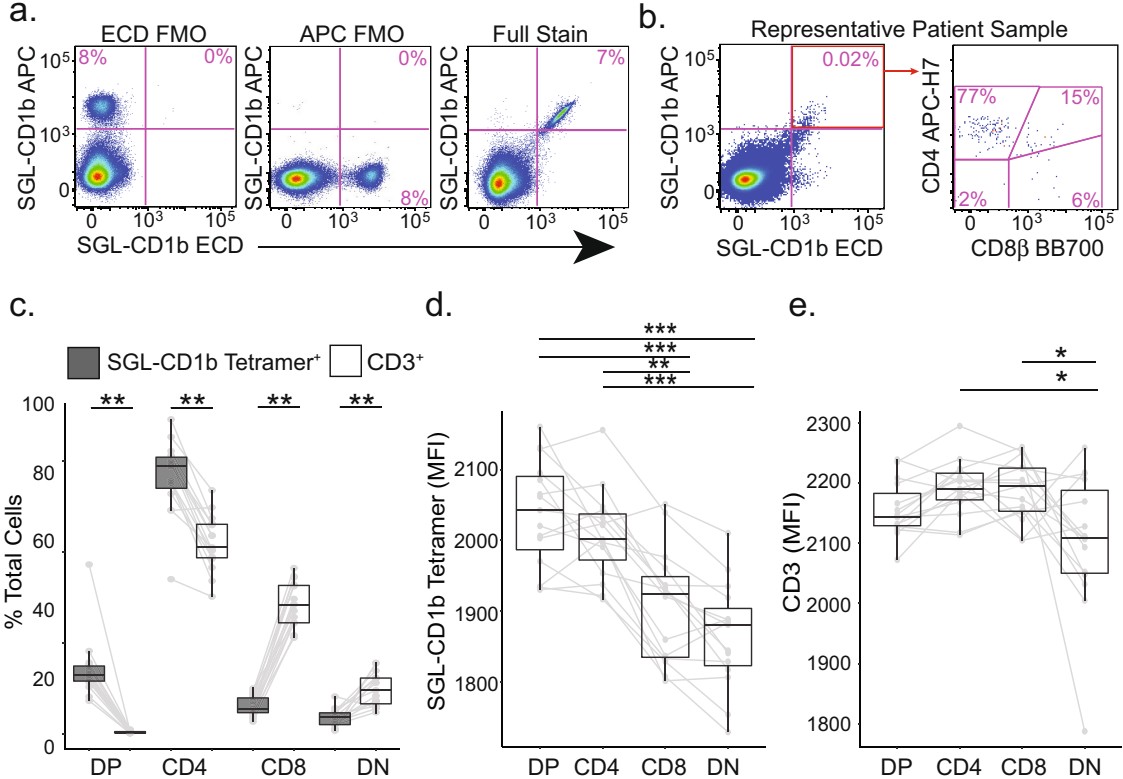

**Fig. 1 Ex vivo co-receptor expression by SGL-specific T cells.** SGL-CD1b tetramers were incorporated into a multi-parameter flow cytometry assay to measure co-receptor expression by SGL-specific T cells. **a** The tetramer-positive gate was defined by a dual tetramer staining with electron coupled dye (ECD) and allophycocyanin (APC)) and 'Fluorescence Minus One' (FMO) negative control (left and center) and a positive control using SGL-specific T cell line (A05) diluted in donor PBMC (right). **b** Representative staining from a South African adolescent blood donor. **c** The co-receptor expression of SGL-CD1b tetramer-positive T cells in the blood was quantified using cryopreserved PBMC obtained from three groups of healthy participants: U.S. controls at low risk for M.tb exposure ($n = 5$), South African adolescents with latent tuberculosis (IGRA-positive, $n = 5$), and South African adolescents without latent tuberculosis (IGRA-negative, $n = 5$). Boxplots depict the minimum and maximum as the smallest and largest number of the dataset, excluding outliers, the median and interquartile range tetramer-positive cells (gray) within each co-receptor group (double positive (DP), CD4, CD8, and double negative (DN), expressed as a percent of total tetramer-positive cells. Each dot represents the percent of one individual, and solid lines connect each individual. The percent of cells in each group was compared to that present in total CD3+ T cells (white). (Two-sided Wilcoxon signed-rank test with Bonferroni Correction, ** = 0.003 or 0.004, $n = 15$). **d** Boxplots depict the minimum and maximum as the smallest and largest number of the dataset, excluding outliers, the median and interquartile range of mean fluorescence intensity (MFI) of SGL-CD1b tetramer-positive cells in each co-receptor group. Each dot represents the MFI of one sample, and solid lines connect each individual. MFI was compared between groups (Two-sided Friedman test with post hoc Dunn test, ***$p < 0.0001$, ** = 0.003, $n = 15$). **e** Boxplots depict the minimum and maximum as the smallest and largest number of the dataset, excluding outliers, the median and interquartile range of mean fluorescence intensity (MFI) of CD3 among tetramer-positive cells in each co-receptor group. Each dot represents the MFI of one sample, and solid lines connect each individual. MFI was compared between groups (Two-sided Friedman test with post hoc Dunn test, *$p = 0.05$, $n = 15$).

**Co-receptors enhance functional avidity of an SGL-specific TCR.** Several factors affect TCR avidity for antigen, including affinity for the antigen-presenting molecule, the level of TCR expression, and the expression of CD4 and CD8 co-receptors on the cell surface[3,28,29]. The aggregate impact these factors have on the binding strength of the TCR in live cells is known as 'functional avidity' due to the multivalent and multifaceted nature of this interaction[3]. Our ex vivo data suggest SGL-specific T cells expressing the CD4 co-receptor may have higher functional avidity than those expressing CD8. To test this hypothesis directly, we examined SGL-specific T cell lines derived from South African adults with LTBI[25]. Virtually all of the SGL-CD1b tetramer staining cells within the A01 T cell line express a heterodimeric CD8αβ co-receptor in contrast to the A05 T cell line, which was strongly biased toward CD4 expression (Supplementary Fig. 2). A05 exhibited 5.5-fold higher SGL-CD1b tetramer MFI compared to A01 ($p = 0.03$) (Supplementary Fig. 2).

We considered the possibility that the observed differences in tetramer MFI between A01 and A05 might be due to differences in their TCRs. Indeed, A01 and A05 each express a distinct TCR, which may have different affinities for the CD1b-SGL complex independent of co-receptor (Table 1). To address this, we cloned and transduced polyclonal T cells isolated from a blood bank donor with an A05 TCR expression construct (Fig. 2a). The constant region of the TCR in the expression construct was replaced with a murine TCR constant region to allow quantification of exogenous TCR expression. Among successfully transduced T cells, as demonstrated by expression of the exogenous TCR and defined by staining with anti-murine TCR-β chain constant region (mTCRBC), 88% also co-stained with SGL-CD1b tetramer, revealing that TCR expression is sufficient to confer SGL-CD1b tetramer binding (Fig. 2a). To further demonstrate the specificity of staining using SGL-CD1b tetramer we also transduced a TCR that is specific for the mycobacterial glycolipid

GMM into Jurkat cells[30] (Supplementary Fig. 3). These GMM-specific Jurkat cells avidly bind GMM-CD1b tetramer, whereas they largely fail to bind SGL-CD1b tetramer and mock-loaded CD1b tetramer (Supplementary Fig. 3). We attempted to also express the A01 TCR in primary T cells, but exceptionally high levels of TCR expression were required for successful tetramer staining. These data support our hypothesis that the difference in tetramer staining between A01 and A05 may be largely due to their TCRs. However, the use of transduced polyclonal T cells

containing CD4, CD8, and DN populations allowed us to specifically study the role of co-receptor independent of the TCR.

Among T cells transduced with the A05 TCR, those also expressing CD4 stained with the highest tetramer MFI, followed by CD8 and DN T cells ($p < 0.0001$) (Fig. 2b). This relative hierarchy was also noted within mTCRBC suggesting that the level of exogenous TCR expression post-transduction also varies based on co-receptor expression ($p < 0.0001$) (Fig. 2b). To explore whether these findings were generalizable, we repeated these experiments using a CD1b-restricted TCR specific for GMM[30]. We again observed a hierarchy in which transduced CD4+ T cells stained with the highest MFI and expressed the highest levels of CD3ε followed by CD8 and DN T cells (Supplementary Fig. 4). We subjected the T cells transduced with the A05 TCR to limiting dilution cloning in an attempt to generate a panel of clones with equivalent levels of exogenous TCR expression. However, we noted that the CD4 clones consistently stained at a higher MFI with SGL-CD1b tetramer than CD8 clones ($p = 0.019$) and also expressed higher levels of mTCRBC ($p = 0.005$) (Fig. 2c). These data suggest that CD4 T cells may intrinsically express higher levels of exogenous TCR than CD8 T cells or possess an intrinsic advantage in their ability to be transduced with lentiviral vectors. Finally, we assessed differences in T cell activation between CD4

**Table 1 TCRs expressed by SGL-specific T Cell Lines.**

| Lines | TCR | Variable | CDR3 | Joining |
|-------|-----|----------|------|---------|
| A01 | α | TRAV8-6 | CAVKAGYSSASKIIF | TRAJ3 |
| | β | TRBV12-4 | CASKGKQGPEQFF | TRBJ2-1 |
| A05 | α | TRAV13-2 | CAEVLGGTSYGKLTF | TRAJ52 |
| | β | TRBV20-1 | CSASGRRGYGYTF | TRBJ1-2 |

Clonality was determined by staining with antibodies targeting human TCR-β variable genes (IOTest BetaMark Kit, Beckman Coulter) or high-throughput sequencing (ImmunoSEQ, Adaptive Biotechnologies). The full-length TCR sequence was determined by template-switched PCR after sorting tetramer-positive T cells. The T cell receptor gene usage and CDR3 region amino acid sequence is summarized here. Variable and joining gene segment names were assigned using the International ImMunoGeneTics (IMGT) Database.

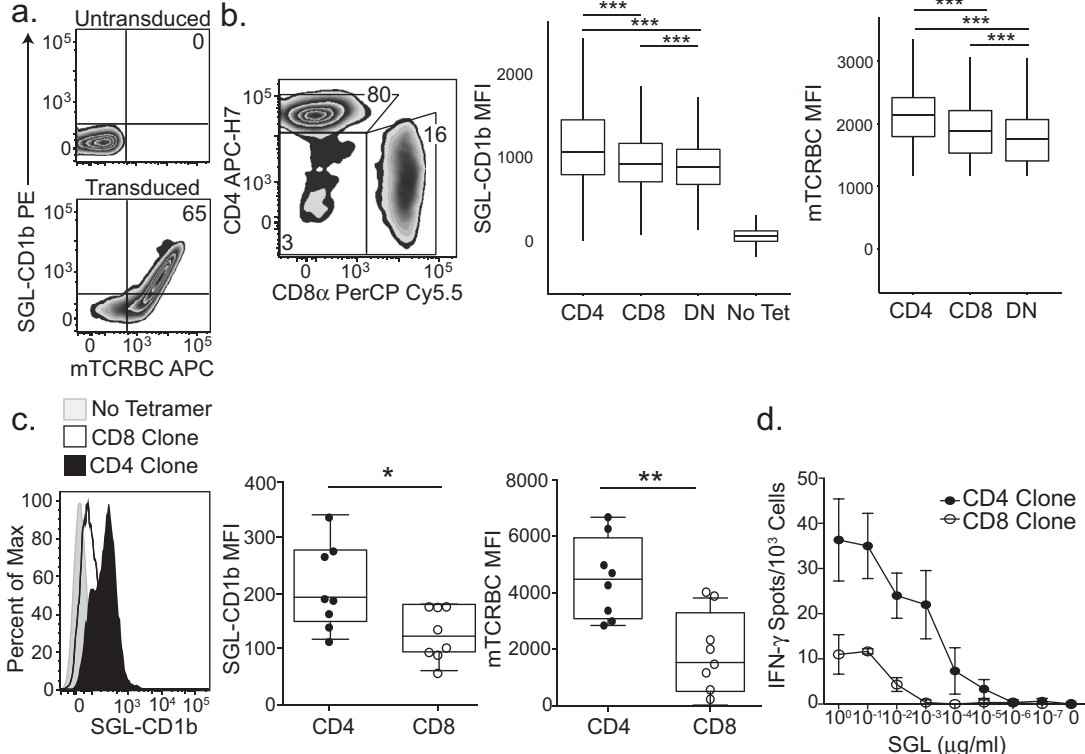

**Fig. 2 Co-receptors enhance functional avidity of an SGL-specific TCR. a** Primary T cells were transduced with the TCR from the A05 T cell line using a lentiviral vector. Among primary T cells that are not transduced with the exogenous TCR, 0% stain with SGL-CD1b tetramers or a mTCRBC antibody (top). After transduction with the TCR, the cells stain with both the SGL-CD1b tetramer and mTCRBC antibody (bottom). **b** Flow plot depicts the percent of transduced T cells that are CD4, CD8, or DN. Boxplot depicts the median and interquartile range of the SGL-CD1b or mTCRBC MFI of each CD4, CD8, and DN T cell that was transduced with the A05 TCR (Two-way ANOVA, post hoc Dunn test, ***$p < 0.0001$, $n = 86,520$). Data are representative of two independent rounds of primary T cell transduction with the TCR of the A05 T cell line. **c** Single-cell clones were generated from the primary T cell transductants by limiting dilution cloning. A representative CD4 (black) and CD8 (white) T cell clone stained with SGL-CD1b tetramer is compared (left). A no tetramer control is also represented (gray) (left). Boxplots depict the minimum and maximum as the smallest and largest number of the dataset, excluding outliers, the median and interquartile range of SGL-CD1b MFI of CD4 (black) and CD8 (white) clones (Two-sided Mann–Whitney test, $p = 0.019$, $n = 16$). **d** Antigen-specific activation of CD4 (black) and CD8 (white) T cell clones was measured using IFN-γ ELISPOT. Error bars represent standard deviation and the center of the error bars is the mean of triplicate wells. SGL was serially diluted log-fold from 1 μg/ml to $10^{-7}$ μg/ml. Data are representative of two independent experiments.

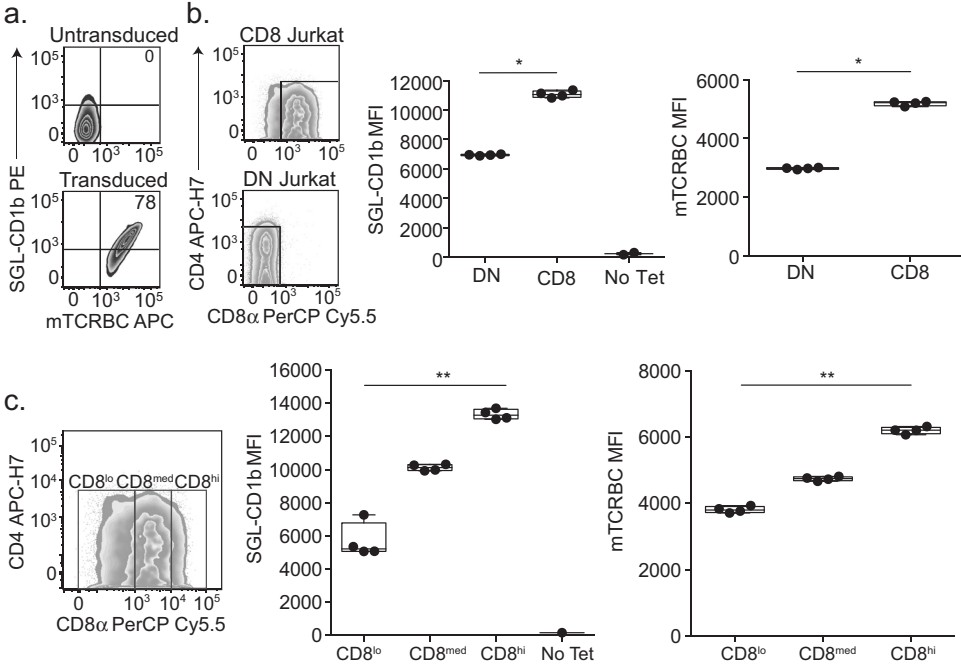

**Fig. 3 CD8 is sufficient to enhance functional avidity of an SGL-specific TCR. a** DN and CD8 Jurkat cell lines were transduced with the TCR from the A05 T cell line. Untransduced CD8 Jurkat cells do not stain with SGL-CD1b tetramer or a murine T cell receptor chain constant region (mTCRBC) specific antibody (top). After transduction with the TCR, the cells stain with both the SGL-CD1b tetramer and mTCRBC antibody (bottom). **b** CD8 and DN Jurkat cells express a gradient of TCR co-receptor expression. Gates enclose true "CD8" (left, top) and "DN" (left, bottom) cells that express only CD8 or no co-receptors, respectively. Of note, the Jurkat cells that express CD8 display a gradient of expression, and the DN Jurkat cells exhibit low levels of CD4 co-receptor expression. The distributions of SGL-CD1b MFI of the DN and CD8 Jurkat cells were then compared (Two-sided Mann–Whitney, $p = 0.03$, $n = 4$). A no tetramer control is also represented (right). The distributions of mTCRBC MFI of the DN and CD8 Jurkat cells were also compared (right) (Mann–Whitney, $p = 0.03$, $n = 4$). Boxplots depict the minimum and maximum as the smallest and largest number of the dataset, excluding outliers, the median and interquartile range of MFI. **c** The distributions of SGL-CD1b MFI of CD8hi (MFI of $10^4$–$10^5$), CD8med (MFI of $10^3$–$10^4$), and CD8lo (MFI of 0–$10^3$) Jurkat cells were assessed (Two-sided Friedman Test with post hoc Dunn Test, $p = 0.005$, $n = 3$). The distributions of mTCRBC MFI of CD8hi, CD8med, and CD8lo Jurkat cells were then compared (right) (Two-sided Friedman Test with post hoc Dunn Test, $p = 0.005$, $n = 3$). Boxplots depict the minimum and maximum as the smallest and largest number of the dataset, excluding outliers, the median and interquartile range of fluorescence intensity. Data are representative of three independent experiments.

and CD8 T cells transduced with the A05 TCR by measuring IFN-γ production in the presence of titrating amounts of SGL. An A05-CD4 T cell clone displayed an $EC_{50}$ of 0.0006 μg/ml, which was 100-fold lower than an A05-CD8 T cell clone (Fig. 2d). Notably, the clones used in this assay also exhibit differences in SGL-CD1b tetramer bindings and TCR expression (Fig. 2c). Taken together, these data demonstrate that compared to CD8 T cells, CD4 T cells exhibit an increase in tetramer binding and TCR expression, which confers enhanced sensitivity to activation at limiting antigen concentrations, despite expressing the same TCR.

**CD8 is sufficient to enhance functional avidity of an SGL-specific TCR.** Primary T cells expressing CD4, CD8, or neither co-receptor exhibit other biological differences besides their co-receptor that might affect functional avidity for SGL-CD1b complex[31]. To address this, we transduced the A05 TCR into a DN Jurkat T cell line and an isogenic T cell line that had been stably transduced to express CD8αα. As with primary T cells, the A05 TCR was sufficient to confer SGL-CD1b tetramer staining in Jurkat T cells (Fig. 3a). The SGL-CD1b tetramer MFI was 1.59-fold higher among CD8αα expressing Jurkat T cells compared to DN T cells ($p = 0.03$) (Fig. 3b). Notably, mTCRBC expression was also 1.74-fold higher among CD8αα compared to DN T cells, indicating higher levels of A05 TCR expression ($p = 0.03$) (Fig. 3b). To investigate this further, we analyzed SGL-CD1b MFI

after stratifying CD8αα expression into logarithmic tertiles (Fig. 3c, left). The CD8hi Jurkat cells had the highest MFI, which was 1.32-fold higher than CD8med Jurkats and 2.35-fold higher than CD8lo Jurkats ($p = 0.0002$) (Fig. 3c, middle). When we compared the level of mTCRBC expression, we found that the CD8hi Jurkat cells again had the highest MFI, followed by the CD8med and CD8lo Jurkats ($p = 0.0002$) (Fig. 3c, right). These data support our hypothesis that co-receptor expression is sufficient to increase functional avidity for SGL-CD1b tetramer, and one mechanism by which this occurs is by increasing the expression of the TCR complex at the cell surface.

**Glycolipid-specific T cells express a diverse TCR repertoire.** To profile CD4 and CD8 glycolipid-specific T cells directly ex vivo, we sorted single cells from four South African adults with newly diagnosed active tuberculosis using SGL-CD1b and GMM-CD1b tetramers[32]. In this experiment, we included GMM-specific T cells and T cell lines with known TCRs as controls to verify our tetramer-sorting strategy. Gates defining CD1b tetramer-positive cells were set based on control samples not stained with the tetramers as well as SGL- and GMM-specific T cell lines (Supplementary Fig. 1). To validate our sorting strategy, we first assessed the TCRs recovered from the T cell lines with a known TCR. We note that the V genes, J genes, and CDR3s recovered from these wells match the known TCRs (Source Data). In addition, as several studies highlighted the enrichment of

TRAV1-2 among GMM-specific TCRs we next compared the frequency of TRAV1-2 among GMM-CD1b tetramer sorted cells and bulk T cells from that same individual to verify the tetramer sorting strategy reliably isolated lipid antigen-specific T cells[30,33,34]. We found that 10% of recovered TCRs express TRAV1-2, compared to 2.9% of bulk T cells ($p = 0.0005$) (Supplementary Fig. 5, Supplementary Table 3). We were also able to identify TCR V genes that have been described within diverse GMM-specific TCR repertoires, such as TRAV17, TRAV8, TRAV12, and TRAV13 (Supplementary Fig. 5)[30,33,34].

As relatively little is known about SGL-specific TCRs, we first compared the recovered TCR variable genes from cells sorted using the SGL-CD1b tetramer to the TCRs from our A01 and A05 T cell lines (Table 1). 4.7% of recovered TCRs from SGL-CD1b tetramer-sorted cells used TRAV8-6 (A01) and 1.6% used TRAV13-2 (A05), compared to 2.1% and 1.0% of bulk T cells, respectively ($p = 0.053$ and $p = 0.35$, respectively) (Supplementary Fig. 5, Supplementary Table 3). We also found that ~6.2% of TCRs recovered from SGL-CD1b tetramer-sorted cells used TRAV21 and 6.2% used TRAV19, compared to 2.9% and 4.2% of bulk T cells, respectively ($p = 0.031$ and $p = 0.26$, respectively) (Supplementary Fig. 5, Supplementary Table 3). These data reveal that SGL-specific T cells isolated and examined directly ex vivo express diverse TCRs that likely have varying functional avidities for CD1b.

Next, we examined the TCR-α chain gene usage of CD4 and CD8 SGL-specific T cells to determine whether the TCR repertoires were distinct between these subpopulations ($n = 38$ and $n = 43$, respectively). Overall, 30% of the V genes we detected were found in both CD4 and CD8 SGL-specific T cells (Supplementary Fig. 6). Of the genes that were found in more than one TCR, TRAV21 and TRAV12-2 were detected preferentially among CD8 SGL-specific T cells ($p = 0.06$ and $p = 0.12$, respectively) (Supplementary Fig. 6). We also found that TRAV2, TRAV29, and TRAV9-2 were detected preferentially among CD4 SGL-specific T cells ($p = 0.10$, $p = 0.21$, $p = 0.21$, respectively) (Supplementary Fig. 6). These data suggest that the TCR repertoire of CD4 and CD8 SGL-specific T cells may be distinct.

**CD4 and CD8 identify functionally distinct compartments among glycolipid-specific T cells**. To examine whether the expression of CD4 or CD8 co-receptors was associated with different functional profiles, we used targeted amplification of 23 genes to examine expression of these transcripts by single GMM- and SGL-specific T cells ($n = 223$) and tetramer-negative T cells ($n = 43$) in two individuals with active tuberculosis. We also included T cell lines with known functional profiles as positive controls ($n = 6$)[35] (Supplementary Tables 4, 5). We performed a supervised analysis focusing on cells in which either CD4 or CD8 transcript was detected (104 of 272 total cells). We first utilized hierarchical clustering to visualize the overall diversity as well as the relationships between CD4+ and CD8 + CD1b-restricted T cells. T cells expressing the CD4 and CD8 co-receptor were spatially separated, suggesting that they express unique transcriptional profiles (Fig. 4a). Of note, we detected expression of all master transcription factors that regulate $T_H1$ (TBET), $T_H2$ (GATA3), $T_H17$ (RORC), and $T_{reg}$ (FOXP3) functional profiles, indicating previously undescribed functional diversity among CD1b-restricted T cells (Fig. 4a). Of note, MKI67 was expressed in 14.2% of CD1b-restricted T cells that were included in our final analysis and 0% off tetramer-negative T cells, which suggests that CD1b-restricted T cells were proliferating at the time of sample collection and that Ki-67 expression is not generally expressed in peripheral blood T cells in individuals with active TB ($p = 0.12$) (Fig. 4a).

We next sought to identify the differentially expressed genes (DEGs) driving the differences between CD4 and CD8 CD1b-restricted T cells (Fig. 4b–e). We found that TBET and PRF1 were enriched in CD8 T cells compared to CD4 T cells, and noted a trend toward the enrichment of EOMES among CD8 T cells, suggesting cytotoxic phenotypes among CD8 mycolipid-restricted T cells (Fig. 4b–d, Supplementary Table 4) ($p = 0.02$, 0.04, 0.08, respectively). We also detected enrichment of the $T_{FH}$ transcription factor BCL6 among CD4 T cells, suggesting functional heterogeneity among CD4 CD1b-restricted T cells (Fig. 4e, Supplementary Table 4) ($p = 0.02$).

These data were complemented by comparing the functional profiles of our in vitro-derived T cell lines, A01 and A05, which express the CD8 and CD4 co-receptor, respectively (Supplementary Fig. 2). We profiled cytokine expression using a previously optimized intracellular cytokine staining panel in the presence of SGL and K562 antigen-presenting cells that have been stably transfected to express CD1b (K562-CD1b) or empty vector (K562-EV)[36,37]. The majority of cells from both T cell lines expressed IFN-γ, TNF, and IL-2 in the presence of SGL (Fig. 5a). In contrast, the T cell lines diverged in the expression of CD40L and CD107a, which are markers of B cell help and degranulation, respectively ($p = 0.03$ and $p = 0.03$, respectively) (Fig. 5a). A01 also secreted 10-fold more granzyme B than A05 and specifically lysed 67% of target cells in the presence of antigen, compared to no significant lysis in the absence of antigen ($p = 0.03$) (Fig. 5b, c, and d). By comparison, A05 did not exhibit cytolytic activity in the presence of antigen above the condition with no antigen present ($p = 0.4$) (Fig. 5c, d). Taken together, these data support the conclusions of our targeted transcriptional profiling experiments and show that SGL-specific T cells expressing CD4 generally express cytokines that align with a T-helper phenotype, while those expressing CD8 have a cytotoxic effector phenotype.

**Discussion**

In summary, we have shown that SGL-specific T cells analyzed directly ex vivo exhibit a bias towards CD4 expression, and our data reveal that CD4 SGL-specific T cells have a higher functional avidity than CD8 SGL-specific T cells. Whether we examined ex vivo, in vitro-derived, or transduced SGL-specific T cells, we observed a consistent hierarchy in which CD4 T cells bind SGL-CD1b tetramer with a higher fluorescence intensity and have the lowest activation thresholds, followed by CD8 T cells, and DN T cells. In fact, our data show that the surface expression of a TCR co-receptor is associated with increased levels of TCR expression, revealing a potential mechanism for the increase in activation and functional tetramer avidity we observed. Finally, we examined the ex vivo transcriptional profiles of SGL-specific T cells in patients with active tuberculosis and found canonical transcription factor lineages among CD4 and CD8 CD1b-restricted T cells. These data significantly advance our understanding of how co-receptors modulate the recognition of lipid antigens and reveal diverse functional profiles of CD1b-restricted T cells in humans.

A previous report found that the CD4 co-receptor is not involved in lipid antigen recognition by CD1b-restricted T cells[38]. This study used blocking antibodies targeting the interaction between CD4 and MHC-II, which may not have reliably predicted the effect on CD1b. Further, the use of co-receptor blocking antibodies can have pleiotropic effects, including paradoxical activation, of T cells[39]. To avoid this limitation, we adopted a molecular approach to this question, and our results uncover a role for TCR co-receptors in lipid-specific T cell activation. Previous reports have shown that the CD4 co-receptor

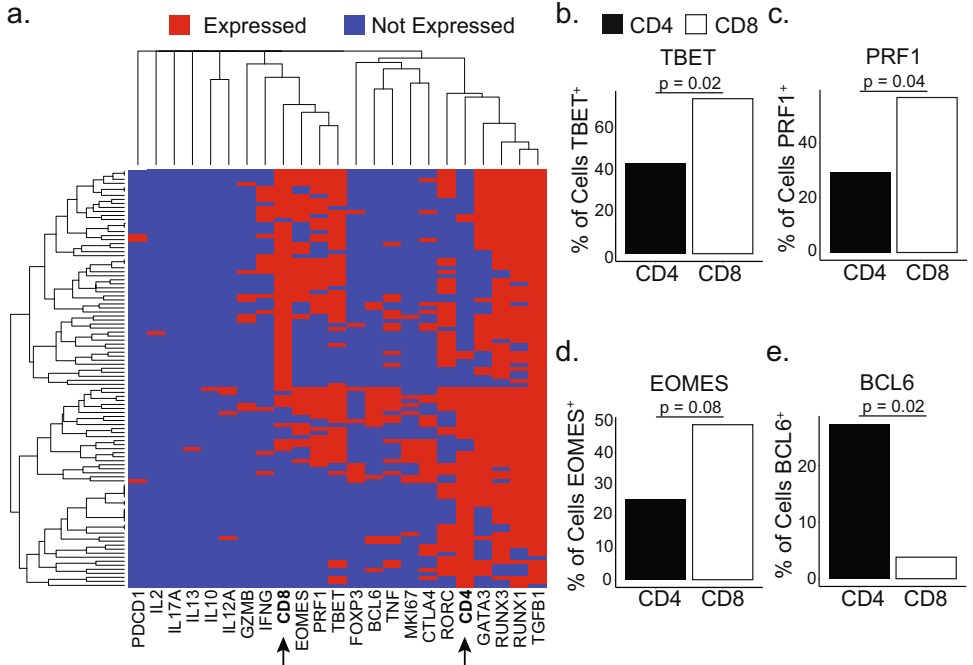

**Fig. 4 Gene expression differences between ex vivo CD4 and CD8 CD1b-restricted T cells. a** Dendrograms on x and y axis summarize similarity between cells (y) or genes (x), where the distance is proportional to the level of dissimilarity. This relationship was computed with complete linkage hierarchical clustering using a binary distance metric. CD4 and CD8 genes on the *X*-axis have been bolded for emphasis. **b–e** Bar charts summarize the percentage of CD4 (black) and CD8 (white) T cells expressing *TBET* (**b**), *PRF1* (**c**), *EOMES* (**d**), and *BCL6* (**e**) (n = 104) (Two-sided Fisher's exact test, Benjamini-Hochberg corrected). Summary of statistical tests is included in Supplementary Table 4.

enhances iNKT cell activation, which the authors posit was through direct binding of CD4 to CD1d[17,40]. Our data extend these studies by revealing a potential molecular mechanism in the form of stabilized TCR expression at the surface of CD1b-restricted T cells, which would not have emerged from experiments using blocking antibodies.

In addition to increasing TCR expression, there may be other potential mechanisms that contribute to increasing the functional avidity of CD1b-restricted T cells, and these mechanisms are not mutually exclusive. First, TCR co-receptors may physically bind to CD1b and augment TCR affinity[41]. However, due to the structural differences between MHC-II and CD1b, putative binding sites for this interaction are not obvious. Second, precise spatial control of the immunologic synapse is important for tuning TCR sensitivity to antigen, possibly by controlling the level of cholesterol in lipid rafts[42–44]. Third, the rate of recycling in endosomes and activation state of T cells can influence TCR stability at the cell surface[45,46]. Due to the multifaceted nature of TCR engagement in live cells, the mechanism for the increase in functional avidity we report here is representative of in vivo antigen recognition and may be unrelated to alterations in the affinity between TCR and CD1b that could be measured using standard biophysical approaches[47].

Our data have important implications for antigen recognition in vivo. Previous studies have highlighted that subtle differences within the lipid tail can significantly influence TCR affinity for antigen, which translates into differences in function[25,48–50]. Our data suggest that co-receptor expression must also be factored into this calculation as our single-cell analyses of SGL- and GMM-specific T cells demonstrate that glycolipid-specific T cells that express CD4 and CD8 co-receptors are functionally distinct. Indeed, our data suggest that these subsets may align closely with peptide-specific T cells, rather than iNKT or MAIT cells[51]. Our

data reveal previously unappreciated functional diversity of CD1b-restricted T cells as we detected expression of *TBET*, *GATA3*, *RORC*, and *FOXP3* among circulating CD1b-restricted T cells ex vivo. Mycobacterial glycolipid-specific T cells that express the CD8 co-receptor are broadly cytotoxic T cells. This is consistent with the original description of an SGL-specific T cell clone, which produced IFN-γ in the presence of M.tb infected cells and reduced the bacterial burden of M.tb in culture[24]. Conversely, T cells that express the CD4 co-receptor are more diverse. At present, the only transcription factor that has been evaluated in CD1b-restricted T cells is PLZF, the transcription factor that regulates innate-like phenotypes in iNKT and MAIT cells[52–54]. In a humanized transgenic mouse model, CD1b-restricted T cells did not express PLZF, which indicates that CD1b-restricted T cells may have a transcriptional landscape that is similar to MHC-restricted T cells[55]. Our findings that CD4 and CD8 CD1b-restricted T cells share major transcriptional pathways with MHC-restricted T cells is consistent with this model[51]. However, we were unable to detect an enrichment of *RUNX3* in CD8 CD1b-restricted T cells, highlighting that there may also exist transcriptional pathways that are unique to CD1b-restricted T cells[56] (Supplementary Table 4).

Together, our data highlight the importance of considering co-receptor expression by CD1b-restricted T cells when evaluating lipid-containing vaccines in clinical and pre-clinical models as CD4 T cells may be more likely to be activated in the context of infection, possibly as a result of enhanced functional avidity. Lipid-based vaccines have shown promise in pre-clinical animal models[57]. CD4 or CD8 CD1b-restricted T cells could be differentially targeted by the adjuvant or delivery platform when 'tuning' the chemical structure of the antigen to modify TCR avidity and immunogenicity of these vaccine strategies.

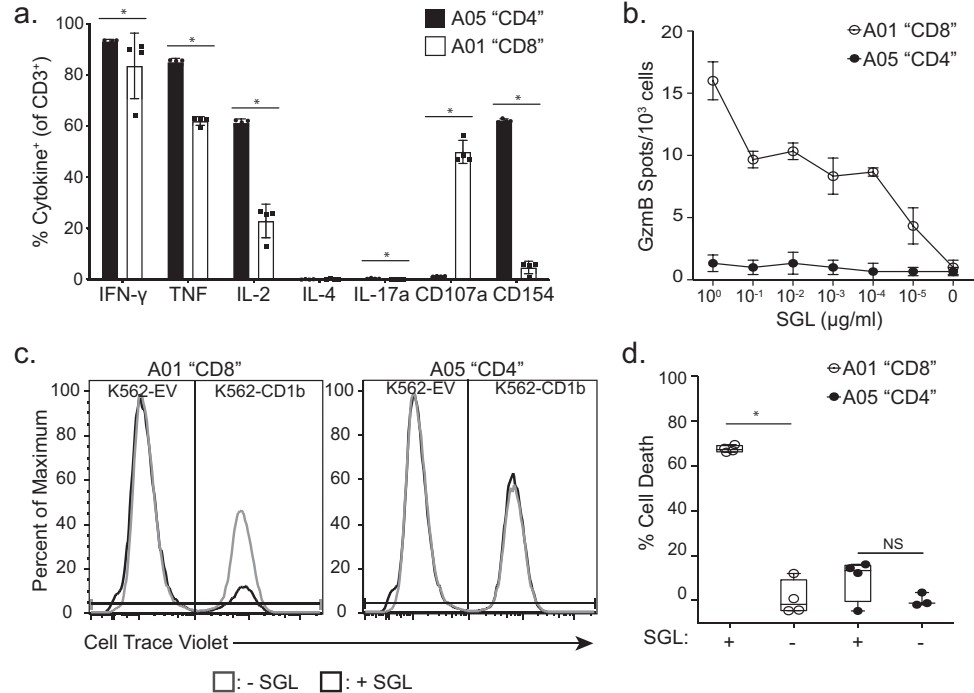

**Fig. 5 Functional profiles of the A01 and A05 T cell lines. a** Cytokine production by the A01 (white) and A05 (black) T cell lines was measured using intracellular cytokine staining after co-incubation with K562-CD1b or K562-EV antigen-presenting cells and SGL antigen for 6 h. Data are expressed as a percent of T cells that express the cytokine as a percentage of total CD3+ T cells. Percentages are background corrected by subtracting the percentage of T cells that express cytokine after co-incubation with K562-EV antigen-presenting cells and SGL for 6 h. Error bars represent the mean and standard deviation of four technical replicates. Dots represent the value from one replicate. (*$p = 0.03$, Two-sided Mann–Whitney, $n = 8$). Data are representative of at least two independent experiments. **b** Granzyme B (GzmB) production from 1000 T cells from the A01 (white) and A05 (black) T cell lines after co-incubation with K562-CD1b antigen-presenting cells and SGL antigen. Cells were cultured overnight and GzmB production was assessed using GzmB ELISPOT. Antigen concentration ranged from 1 µg/ml to $10^{-5}$ µg/ml in log-fold dilutions. Error bars represent standard deviation and the center of the error bars is the mean of triplicate wells. Data are representative of two independent experiments. **c** Cytotoxicity was assayed by co-incubating A01 (left) and A05 (right) T cell lines with K562-EV and K562-CD1b antigen-presenting cells labeled "low" and "high" with Cell Trace Violet, respectively. Co-cultures were incubated in the presence (black) or absence (gray) of SGL antigen. We defined specific lysis as the ability of a T cell line to specifically reduce the K562-CD1b cell population in the presence of lipid, while leaving the K562-EV population and the K562-CD1b without pulsed antigen intact after co-incubation. **d** Cytotoxicity was quantified by calculating the percent of cell number reduction of K562-CD1b cells in the experimental conditions compared to the percentage of K562-CD1b cells in culture in a condition with no T cells and no lipid antigen (Supplementary Fig. 7) (% Cell Death). Percentages were calculated for the A01 (white) and A05 (black) T cell lines in the presence or absence of SGL. Error bars represent the mean and standard deviation of four technical replicates (*$p = 0.03$, Two-sided Mann–Whitney, $n = 8$). Boxplots depict the minimum and maximum as the smallest and largest number of the dataset, excluding outliers, the median and interquartile range of % Cell Death measurements. Data are representative of at least two independent experiments.

## Methods

**Clinical cohorts**. For ex vivo analysis of SGL-CD1b tetramer-positive cells, we studied two cohorts of healthy participants. First, U.S. healthy controls were recruited and enrolled at the Seattle HIV Vaccine Trials Unit as part of a cohort of healthy adults who provided blood samples for developing and testing new assays. Peripheral blood mononuclear cells (PBMC) were collected by leukapheresis from five HIV-seronegative individuals with a known T cell response to CMV were used here. Second, we studied a subset of 6363 South African adolescents who were enrolled into a study that aimed to determine the incidence and prevalence of tuberculosis infection and disease in South Africa[26]. Adolescents aged 12- to 18-years-old were enrolled at eleven high schools in the Worcester region of the Western Cape of South Africa. Participants were screened for the presence of LTBI by a tuberculin skin test and/or IFN-γ release assay (IGRA) QuantiFERON-TB GOLD In-Tube (QFTG) (Cellestis Inc.) at study entry. PBMC were isolated from freshly collected heparinized blood via density centrifugation and cryopreserved. For this work, a sample of five M.tb-infected (QFTG+) and five M.tb-uninfected (QFTG−) adolescents were selected based on the availability of PBMC.

We also utilized cryopreserved PBMC samples isolated from adults with a new diagnosis of active tuberculosis also from the Worcester region of the Western Cape of South Africa[32]. Participants were over 18 years of age and HIV-uninfected. All patients had positive sputum smear microscopy and/or positive culture for M.tb. Blood was obtained and PBMC archived prior to or within 7 days of starting standard course anti-tuberculosis treatment, which was provided according to guidelines of the South African National Tuberculosis Programme.

**Ethics statement**. The study protocols were approved by the IRBs of the University of Washington, the Fred Hutchinson Cancer Research Center, and the University of Cape Town. Written informed consent was obtained from all adult participants as well as from the parents and/or legal guardians of the adolescents who participated. In addition, written informed assent was obtained from the adolescents.

**Culture media**. Media (R10) for washing PBMC consisted of RPMI 1640 (Gibco, Waltham, MA) supplemented with 10% fetal calf serum (Hyclone, Logan, UT). Our base T cell media (TCM) consisted of sterile-filtered RPMI 1640 supplemented with 10% fetal calf serum, 100 U/ml Penicillin, 100 mg/ml Streptomycin, 55 mM 2-mercaptoethanol, 0.3X Essential Amino Acids, 60 mM Non-essential Amino Acids, 11 mM HEPES, and 800 mM L-Glutamine (Gibco, Waltham, MA). Our TCM containing human serum (TCM/HS) consisted of sterile-filtered RPMI 1640 supplemented with 10% human serum (derived from healthy donors), 100 U/ml Penicillin, 100 mg/ml Streptomycin, and 400 mM L-Glutamine (Gibco, Waltham, MA). For culture of Jurkat cells, enhanced RPMI was used, which consisted of RPMI 1640 (Gibco, Waltham, MA) supplemented with 10% fetal calf serum (Hyclone, Logan, UT), 100 U/ml Penicillin, 100 mg/ml Streptomycin, and 800 mM L-Glutamine (Gibco, Waltham, MA).

**Generation of SGL- and GMM-loaded tetramers**. Soluble biotinylated CD1b monomers were provided by the National Institutes of Health Tetramer Core Facility (Emory University, Atlanta, GA). The loading protocol for CD1b

monomers was based on previously published loading protocols[25,58]. For SGL-loaded CD1b tetramers, a biologically validated synthetic analog of SGL, referred to as AM Ac₂SGL in James et al.[25], was used for all experiments except those involving single-cell profiling for which we used SGL purified from cell wall extracts from *Mycobacterium tuberculosis*[24,25]. SGL was dried down in a glass tube in a stream of nitrogen and sonicated into a 50 mM sodium citrate buffer at pH 4, containing 0.25% 3-[(3-cholamidopropyl)dimethylammonio]-1-propanesulfonate (CHAPS) (Sigma, St. Louis, MO) for two minutes at 37 °C. For GMM-loaded CD1b tetramers, C₃₂ GMM derived from *Rhodococcus equi* was dried down and sonicated into 50 mM sodium citrate buffer at pH 4, containing 0.25% CHAPS. The sonicate was transferred to a microfuge tube, and 20 μl of CD1b monomer was added and incubated in a 37 °C water bath for 2 hours with vortexing every 30 min. At the end of the incubation, the solution was neutralized to pH 7.4 with 6 μl of 1 M Tris pH 9. After the addition of CD1b, the mixture was incubated in a 37 °C water bath for 2 h. Finally, 10 μl of Streptavidin conjugated to ECD or APC (Life Technologies, Carlsbad, CA) was added in ten aliquots of 1 μl every 10 minutes. The final product was filtered through a SpinX column (Sigma, St. Louis, MO) to remove aggregates and stored at 4 °C until use.

## Tetramer staining and sorting

*Ex vivo analysis.* PBMC were thawed in warm thaw media (R10 with 2 μl/ml Benzonase (Millipore, Billerica, MA) sterile-filtered) and centrifuged at 700 × *g* for 5 min. The supernatant was decanted, and the cells were resuspended in R10 and counted by Trypan Blue (Millipore, Billerica, MA) exclusion. The cells were centrifuged at 700 × *g* for 5 min and plated at a density of 1 million cells per well in a 96-well U-bottom plate. A portion of the PBMC were resuspended in R10 at a density of 2 million cells per ml in 50 ml conicals with the caps lightly in place and rested overnight at 37 °C in humidified incubators supplemented with 5% CO₂. The PBMC in the 96-well plate was washed with FACS buffer (1× phosphate-buffered saline (PBS) (Gibco, Waltham, MA) supplemented with 0.2% bovine serum albumin (BSA) (Sigma, St. Louis, MO) and centrifuged at 700 × *g* for 3 min. Next, the cells were washed twice with PBS and stained with Aqua Live/Dead stain (cat. L34966; Life Technologies, Carlsbad, CA) according to the manufacturer's instructions. Following a 15 min incubation at room temperature, the cells were washed twice in PBS. They were then blocked with human serum (Valley Bio-medical, Winchester, VA) and FACS buffer mixed 1:1 for 10 min at 4 °C. The wells were washed twice with FACS buffer and then resuspended in 50 μl FACS buffer with 1 μl of unloaded CD1b tetramer and 1 μl of each SGL-loaded CD1b tetramer labeled with APC or ECD and incubated at room temperature for 40 min in the dark. After this incubation period, the cells were washed twice with FACS buffer and then labeled with anti-CD3 ECD (1:50 dilution) (cat. IM2705U; Beckman Coulter, Brea, CA), anti-CD4 APC Ax750 (1:50 dilution) (cat. A94685; Beckman Coulter, Brea, CA), and anti-CD8β BB700 (1:10 dilution) (cat. 745761; BD Biosciences, San Jose, CA) antibodies for 30 min at 4 °C. After two final washes in FACS buffer, the cells were fixed in 1% paraformaldehyde (PFA) (Electron Microscopy Sciences, Hatfield, PA) and acquired on a BD LSRFortessa running BD FACSDiva (BD Biosciences, San Jose, CA) equipped with blue (488 nm), green (532 nm), red (628 nm), violet (405 nm), and ultraviolet (355 nm) lasers.

*In vitro-derived T cell lines and transduced T cells.* T cell lines or transductants were plated at one million cells per well in a 96-well U-bottom plate. Cells were washed twice with PBS and resuspended with Live/Dead Fixable Aqua or with Live/Dead Fixable Green Dead Cell Stain Kit (cat. L34966 or cat. L23101, respectively; Life Technologies, Carlsbad, CA) per the manufacturer's instructions. For this step and all subsequent steps, the cells were kept in the dark. Following a 15 min incubation, cells were washed twice with PBS and blocked with human serum (Valley Bio-medical, Winchester, VA) prepared in FACS buffer (1× PBS (Gibco, Waltham, MA) supplemented with 0.2% BSA (Sigma, St. Louis, MO)) mixed 1:1 for 10 min at 4 °C. Cells were then resuspended in 50 μl of FACS buffer containing 1 μl of SGL-loaded CD1b and 1 μl of mock-loaded control CD1b tetramers, then incubated at room temperature for 60 min. The tetramer titers were determined prior to use in the present study. Following a 15 min incubation at room temperature, the cells were washed twice in PBS and then stained with anti-CD3 ECD (1:50 dilution) (cat. IM2705U; Beckman Coulter, Brea, CA), CD4 APC-Ax750 (1:50 dilution) (cat. A94685; Beckman Coulter, Brea, CA) and anti-CD8α PerCP Cy5.5 (1:10 dilution) (cat. 341051; BD Biosciences, San Jose, CA). When T cells transduced with exogenous TCR were used, anti-mouse TCR β chain APC (1:50 dilution) (cat. 553174; BD Biosciences, San Jose, CA) was also included in the antibody cocktail. The optimal titers of all antibodies were determined prior to use. After two final washes in FACS buffer, the cells were fixed in 1% paraformaldehyde (PFA) (Electron Microscopy Sciences, Hatfield, PA) and acquired on a BD LSRFortessa as above.

*Sorting.* The transduced T cells were sorted to purity using a modified version of the tetramer staining method described above. After the antibody stain, the transduced T cells were resuspended in 200 μl of FACS buffer and tetramer-positive T cells were sorted at the UW Department of Immunology Flow Cytometry Core using a FACS Aria II (BD Biosciences, San Jose, CA) cell sorter equipped with blue (488 nm), red (641 nm), and violet (407 nm), lasers. Cells were sorted into 3 ml of TCM in 4 ml FACS tubes.

## TCR cloning and transduction

*Template-switched PCR.* The sequences of the A01 and A05 TCR were determined by a previously described 5′-rapid amplification of cDNA ends (5′RACE) based cloning strategy based on the manufacturer's instructions[59] (Takara Bio, Japan). Briefly, RNA was extracted from T cells and cDNA was synthesized using a dT oligo and template-switch adaptor. Then, customized oligos targeting the TCRα and TCRβ constant region were used to amplify TCR sequence when combined with a universal primer. The oligo sequences are as follows:

TCRα chain primer: 5′-GATTACGCCAAGCTTGTTGCTCCAGGCCACAGCACTGTTGCTC-3′

TCRβ chain primer: 5′-GATTACGCCAAGCTTCCCATTCACCCACCAGCTCAGCTCCACG.3′

The products were cloned into a sequencing vector and sequenced using Sanger sequencing. The variable genes, joining genes, and CDR3 used by the sequences were analyzed using the IMGT Database (Montpellier, France).

*TCR cassette construction.* Codon-optimized A01 and A05 T cell TCR sequences were assembled into an expression cassette. In this construct, the TCR α and TCR β chain constant regions are replaced with modified murine TCR α chain and TCR β chain constant regions (mTCRBC) to facilitate measurement of TCR expression by flow cytometry. It also contains an extra cysteine residue to encourage pairing between exogenous TCR chains. The TCR cassettes were synthesized through Thermo Fisher GeneArt Synthesis service and were then cloned into pRRL.PPT.MP.GFPpre[59,60] using BamHI and SalI restriction enzymes (New England BioLabs, Ipswich, MA). All plasmids were purified using Maxi Prep kits (Qiagen, Hilden, Germany).

*Generation of lentivirus.* Lenti-X HEK293T cells (Clontech, Mountain View, CA) were seeded at 2 million cells per 100 mm tissue culture dish and incubated for 48 h at 37 °C/5% CO₂ in DMEM (Gibco, Waltham, MA) or until cells reached 75% confluency. The medium was replaced 4 h before transfection. Cells were transfected with 10 μg pRRL-TCR plasmid, 5 μg pCI-VSVG envelope plasmid, and 5 μg of a psPAX2 packaging vector (gifted from Dr. Stanley Riddell at Fred Hutchinson Cancer Research Center). Plasmids were mixed with Fugene 6 transfection reagent (Promega, Madison, WI) at a dilution of 1:12 in a total volume of 600 μl. Transfection mixture was added dropwise into the cell culture and incubated overnight in the conditions described above. The medium was then replaced and incubated for an additional 48 h. After this time, 20 μl of supernatant was titered using Lenti-X GoStix (Clontech, Mountain View, CA) per the manufacturer. The supernatant was then harvested every 12 h for a total of three collections. At each collection, cell debris was removed by centrifugation at 700 × *g* for 5 min, and cleaned supernatant was reserved in a 50-ml conical and kept at 4 °C until three collections had been acquired. The supernatant was then incubated overnight with Lenti-X concentrator (Clontech, Mountain View, CA) at a ratio of 1:3. The following day, the supernatant was centrifuged at 1500 × *g* for 45 min at 4 °C. The supernatant was then discarded, and the pelleted virions were resuspended in 300 μl R10 media and stored at −80 °C until further use. Viral preparations were titred for optimal transduction efficiency in Jurkat cells.

*Transduction of Jurkat cells.* CD8-expressing and DN Jurkat cells were seeded at 1 million cells per well in a 48-well plate in enhanced RPMI. Jurkat cells were generously provided by Dr. Thomas Blankenstein. The same day, the Jurkat cells were transduced with TCR lentivirus at an estimated multiplicity of infection (MOI) of 5 with 1 μl of polybrene at a final concentration of 4 μg/ml (Sigma, St. Louis, MO). Cells and virus were incubated for 4 h at 37 °C/5% CO₂ and washed with PBS (Gibco, Waltham, MA) to remove excess virus. Jurkat cells were maintained in culture for 1 week using enhanced RPMI and screened for TCR expression[59] using an anti-mouse TCR β chain (mTCRBC) APC antibody (1:50 dilution) (cat. 553174; BD Biosciences, San Jose, CA) and tetramer staining as described above.

*Transduction of primary T cells.* Cryopreserved PBMC from an anonymous blood bank donor (Bloodworks Inc, Seattle, WA) was thawed as described above and enumerated using trypan blue exclusion. T cells were isolated from PBMC using magnet-activated cell sorting (MACS) using the Pan T cell Isolation Kit (negative selection) according to the manufacturer's instructions (Miltenyi Biotec, Germany). Following separation, T cells are enumerated using Trypan Blue exclusion and co-incubated with CD3/CD28 Human T cell Activating Dynabeads (ThermoFisher) at a 3:1 bead:T cell ratio in a total volume of 1 ml TCM supplemented with 50 units/ml recombinant IL-2 in a 48-well plate. On day 2, lentiviral stock and 4 μg/ml polybrene was added to culture. Viral titer is determined prior to use. Following viral addition to the well, the 48-well plate is then centrifuged at 679 × *g* for 90 min at 32 °C. The cells are then incubated at 37 °C/5% CO₂ for 48 h. On day 4, the cells are harvested and transferred to a 4 ml FACS tube and rinsed with 3 ml of TCM. The cells are then centrifuged at 700 × *g* for 5 min to remove the polybrene from the cells. Following centrifugation, the cells are resuspended in TCM supplemented with rIL-2 as described above and returned to the original plate. The cells are then incubated for 72 h at 37 °C/5% CO₂. On day 7, the cells are harvested and moved to a 4 ml FACS tube and incubated on a magnetic stand for 10 min at room temperature to remove the Dynabeads. The media, which contains the activated T cells,

was then removed and added to a 15 ml conical. Fresh media is then added to the 4 ml FACS tube to rinse the beads, and the 4 ml FACS tube is then returned to the magnetic stand and incubated for 5 min at room temperature. The media is then removed from the 4 ml FACS tube and added to the 15 ml conical. Cells are then centrifuged at $700 \times g$ for 5 min and resuspended in fresh TCM supplemented with rIL-2 and returned to the 48-well plate. Cells are then incubated for 4 days at 37 °C/ 5% $CO_2$ and maintained as necessary. On day 12, the cells are then assayed to quantify T cell transduction by tetramer staining as described above.

## T cell culture and analysis

*Rapid expansion method.* Further expansion of the T cells transduced with exogenous TCR was performed using a modified version of a previously established rapid expansion protocol[61]. Briefly, 100,000 T cells were mixed with 5 million irradiated EBV-transformed B cells and 25 million irradiated PBMC as feeder cells in R10 in T25 tissue culture flasks (Costar, St. Louis, MO) with 25 ml TCM. Anti-CD3 (clone OKT3) was added at a final concentration of 30 ng/ml, and the mixture was incubated overnight at 37 °C/5% $CO_2$. The following day, recombinant human IL-2 (rIL-2) (Prometheus Pharmaceuticals through UWMC Clinical Pharmacy) was added at a final concentration of 50 U/ml. On day 4, the cells were washed twice in TCM to remove the anti-CD3 antibody and resuspended in fresh media supplemented with rIL-2 at 50 U/ml. Half the media was replaced every three days or split into new T25 tissue culture flasks as determined by cell confluency. After 13 days in culture, the transduced T cells were screened by tetramer staining and then frozen on day 14.

*Limiting dilution cloning.* Sorted T cells were washed and resuspended in TCM/HS and plated at 1 or 2 cells per well in each well of a 96-well plate to create single-cell clones. Irradiated PBMC (150,000 cells per well) were added as feeder cells along with PHA (Remel, San Diego, CA) at a final concentration of 1.6 µg/ml. After two days in the culture at 37 °C/5% $CO_2$, 10 µl natural IL-2 (Hemagen, Columbia, MD) was added to each well. Half the media was replaced every two days with TCM/HS and natural IL-2. Cultures were maintained for 14 days and screened for antigen specificity by tetramer staining and IFN-γ ELISPOT.

*Intracellular cytokine staining.* On day 0, cryopreserved T cell lines were thawed, counted, and enumerated using Trypan blue exclusion. T cell lines were rested in TCM overnight as described above and enumerated again on day 1. On day 0, SGL was evaporated to dryness from chloroform-based solvents under a sterile nitrogen stream and then sonicated into media. This lipid suspension was added to 50,000 K562-EV or K562-CD1b cells at 5 mg/ml final concentrations. K562 cells were incubated for 18 h at 37 °C, 5% $CO_2$ to facilitate lipid loading. On day 1, rested T cell lines were split and added to the K562 cells without washing at a final density of 1 million/well. Thus, each T cell line was co-incubated with loaded K562-CD1b and K562-EV antigen-presenting cells. The cell mixture was allowed to incubate for 6 h in the presence of anti-CD28/CD49d (1:25 dilution) (cat. 347690; BD Biosciences, San Jose, CA), brefeldin A at a final concentration of 10 mg/ml (Sigma-Aldrich), GolgiStop containing Monensin (BD Biosciences, San Jose, CA), and anti-CD107a (1:50 dilution) (cat. 561348; BD Biosciences, San Jose, CA), after which EDTA, at a final concentration of 2 mM, was added to disaggregate cells. On day 2, the samples were washed twice in PBS and then stained with Aqua Live/ Dead (cat. L34966; Life Technologies, Carlsbad, CA) prepared according to manufacturer's instructions and incubated for 20 min at room temperature. Live/Dead staining and all steps following were performed in the dark. The cells were washed twice in PBS and then incubated at room temperature for 10 min in 1x FACS Lyse (BD Biosciences, San Jose, CA). Following one wash with FACS buffer, the cells were incubated an additional 10 min in 1x FACS Perm II (BD Biosciences, San Jose, CA) at room temperature. The cells were washed twice in FACS buffer and labeled with antibodies for CD3 (cat. IM2705U; 1:50 dilution) and CD4 (cat. A94685; 1:50 dilution) (Beckman Coulter, Brea, CA), CD8 (cat. 341051; 1:10 dilution), IFN-γ (cat. 560371; 1:50 dilution), IL-2 (cat. 559334; 1:10 dilution), IL-4 (cat. 554486; 1:250 dilution), CD154 (cat. 555701; 1:5 dilution), and TNF (cat. 554512; 1:50 dilution) (BD Biosciences, San Jose, CA) for 30 min at 4 °C. Following two final washes in FACS buffer, the cells were fixed in 1% PFA and acquired on a BD LSRFortessa as above (BD Biosciences, San Jose, CA).

*IFN-γ and granzyme B ELISPOT.* EMD Multiscreen-IP filter plates (Millipore, Billerica, MA) were coated with 1D1K antibody (Mabtech, Sweden) diluted 1:400 in PBS and incubated overnight at 4 °C. T cell lines were rested. One thousand T cells were plated 1:50 with K562-CD1b and K562-EV cells. Lipids antigens were stored in chloroform:methanol (2:1, v-v) at a stock concentration of 1 mg/ml and a temperature of −20 °C. For these experiments, ~4 µl were dried in glass centrifuge tubes under a sterile nitrogen stream. The lipids were sonicated into TCM to obtain a 4 µg/ml suspension and plated with the cells at a final concentration of 1 µg/ml. These cultures were incubated at 37 °C for ~16 h. The following day, the cells were washed twice with sterile water to lyse the cells, and the plates were incubated with the detection antibody 7-B6-1-biotin (Mabtech, Sweden) diluted 1:1000 in PBS + 0.5% FCS and incubated for two hours at room temperature. The cells were then washed five times with PBS and incubated in ExtrAvidin-Alkaline Phosphatase (Sigma, St. Louis, MO) diluted 1:1000 in PBS and incubated for one hour at room temperature. The wells were then washed five times and incubated with BCIP/NBT

substrate (Sigma, St. Louis, MO) for five minutes to develop the membrane. The wells and IFN-γ spots were counted using an ImmunoSpot S6 Core Analyzer (Cellular Technology Limited, Cleveland, OH). The granzyme B ELISPOT followed identical methods with the following exceptions: EMD filter plates were coated with GB10 antibody diluted to 15 µg/ml, GB11-biotin diluted to 1 µg/ml was used as the detection antibody, and the culture incubation was extended to 48 h (Mabtech, Sweden).

*Cytotoxicity assay.* To detect cytotoxic activity, the T cell lines were thawed and rested overnight in TCM as described above. The following day, K562-CD1b and K562-EV were stained with Cell Trace Violet (Invitrogen, Carlsbad, CA) at 10 µM and 0.2 µM, respectively, for 20 min at 37 °C. Staining was quenched by adding 10 ml cold PBS to the staining solution. Stained K562-CD1b and K562-EV cells were then mixed at a 1:1 ratio. This cell mixture was then co-incubated with the T cell lines at an effector to target ratio of 5:1 in a 96-well plate in a total volume of 200 µl of TCM. SGL was added to the culture media at 1 µg/ml, and the cell suspension was centrifuged briefly to pellet the cells. Cells were incubated for 24 h at 37°C/5% $CO_2$. The following day, the cells were washed twice with PBS, and stained with Aqua Live/Dead (cat. L34966; Life Technologies, Carlsbad, CA) prepared according to the manufacturer's instructions, and incubated for 20 min at room temperature. The cells were then washed twice with PBS and then stained with anti-CD3 antibody (1:50 dilution) (cat. IM2705U; Beckman Coulter, Brea, CA) for 30 min at 4 °C in FACS buffer. The cells were then washed twice with FACS buffer and fixed in 1% PFA. Samples were acquired on an LSR Fortessa as above (BD Biosciences, San Jose, CA). We defined "% Cell Death" as the percent change of the proportion of K562-CD1b cells that remain after co-culture compared to the proportion of K562-CD1b cells at baseline. The baseline condition is defined by the mixture of K562-CD1b and K562-EV cells where no T cells or lipid antigen was present in the culture (Supplementary Fig. 7).

**Targeted transcriptional profiling.** Transcriptional profiles and TCR sequences were determined from sorted cells using an established protocol[35]. Briefly, cells were sorted from cryopreserved PBMC and T cell lines as positive controls using CD1b tetramers loaded with GMM or purified SGL into single wells of a 96-well PCR plate (Eppendorf, Hamburg, Germany), which contained 5 µl of sort buffer consisting of (1X Qiagen One-step RT PCR Buffer (Qiagen, Hilden, Germany), 0.1 mM dithiothreitol (Invitrogen, Carlsbad, CA), RNaseOUT (Invitrogen, Carlsbad, CA), and molecular-grade nuclease-free water (Fisher, Hampton, NH)). Cells were centrifuged briefly at $700 \times g$ for 1 min, and frozen at −80 °C overnight. The following day, the reverse transcription reaction was performed using primer pools containing TCRAV and TCRBV primers at 0.06 µM final concentration, TCRA and TCRB constant region primers at 0.3 µM final, and phenotype gene primers at 0.1 µM. Primer sequences are as described in Han et al.[35], with exceptions as noted in Supplementary Tables 5 and 6. Samples then undergo nested amplification in a Mastercycler Nexus Gradient Thermal Cycler (Eppendorf, Hamburg, Germany). The second phase of amplification uses 0.4 units of HotStarTaq DNA polymerase (Qiagen, Hilden, Germany). Lastly, the amplicons were barcoded according to their plate, row, and column location using the barcodes specified in the published protocol[35]. The TCR α chain, TCR β chain, and phenotyping genes were amplified separately at this step. The row and column barcodes were each added at 0.05 µM, and Illumina MiSeq paired-end adapters were used at 0.5 µM. After the amplification steps, all wells were pooled from each plated and cleaned using Ampure XP PCR Cleanup Beads (Beckman Coulter, Brea, CA) per the manufacturer's instructions. Each plate was pooled and the plate libraries were quantified on a Qubit Fluorometer 3.0 using a dsDNA High-sensitivity Kit (Life Technologies, Carlsbad, CA). The plates were then pooled in equal volumes and the pooled library was then diluted to 6 µM and sequenced using a 500-cycle V3 reagent on an Illumina MiSeq (Illumina, San Diego, CA), which yields 25 million paired reads of 250 base pairs, at the Fred Hutchinson Cancer Research Center Genomics Core.

**Transcriptional profiling computational methods.** Raw reads were trimmed and demultiplexed using VDJFasta software[62]. Paired-end reads were assembled by finding a consensus of at least 100 bases. Amplicons smaller than 100 bases were removed. The resulting paired-end reads were then assigned to wells according to barcodes that designate the plate, row, and column. The average read count for this study was ~47,000 reads per well.

To map the TCR reads, a cutoff of >95% sequence identity was established to collapse reads and establish a consensus sequence within each well. All sequences exceeding 95% sequence identity are assumed to derive from the same TCR sequence. The 95% cutoff conservatively ensures all sequences derived from the same transcript would be properly assigned. In addition, TCR assignments that fail to pass a minimum frequency of 0.1 for each well are removed from the dataset as they might be derived from sorted doublets[35]. For this dataset, the median frequency of the dominant TCR frequency in our dataset is 0.79 with a range of 0.313–0.999.

For phenotyping transcripts, the number of reads containing a 95% match to the customized database of transcription factors and cytokine genes are scored. Genes profiled in this study are summarized in Supplementary Tables 5 and 6. The resulting data are then compiled into a.csv file containing the TCR assignment and the number of reads for each phenotype gene for each well. A minimum read count

of five reads per gene was used to define the expression of each gene as established in Han et al.[35].

Data were then input into the R programming environment and further processed and formatted. The resulting matrix was then analyzed using hierarchical clustering to visualize relationships between functional profiles expressed by T cells where CD4 or CD8 were detected ($n = 104$ of 272 total sorted cells). For T cells where both CD4 and CD8 were detected, these were coded as "CD4", as most exhibited a bias toward CD4 expression, and biologically CD4 T cells that also express CD8a are likely to be activated CD4 T cells[63]. Fisher's Exact Tests were used to determine which genes were significantly enriched among CD4 or CD8 T cells and p-values were adjusted for multiple comparisons using the Benjamini-Hochberg method. Other data visualization and statistical tests were performed in Graph Pad Prism (v6) (San Diego, CA).

**Computational and statistical methods.** Initial compensation, gating, and quality assessment of flow cytometry data was performed using FlowJo version 9 (FlowJo, TreeStar Inc, Ashland OR). Flow cytometry data processing and MFI extraction were performed using the OpenCyto framework in the R programming environment[64]. Statistical tests are described in the Figure and Table legends. Categorical variables were analyzed using a Fisher's exact test. When two continuous variables were analyzed, a Student's t-test was used when data were normally distributed, and a Mann–Whitney test was used when normality assumption could not be met. When more than two continuous variables were analyzed, a two-way analysis of variance (ANOVA) was used when data were normally distributed, and a Kruskal–Wallis test was performed when a normal distribution could not be assumed. A Wilcoxon Signed Rank test or a Friedman test was used when the continuous variables were not independent for two groups and three or more groups, respectively. Post hoc Dunn tests were performed after ANOVA, Kruskal–Wallis, and Friedman tests to determine which group(s) were different. When multiple hypotheses were tested, p-values were adjusted using the Bonferroni method. These tests were conducted in R (v3.8.5) or Graph Pad Prism version 6 (GraphPad, GSL Biotech, San Diego, CA).

**Reporting summary.** Further information on research design is available in the Nature Research Reporting Summary linked to this article.

## Data availability
Annotated data, as well as the code used to generate these data and figures are available at: https://github.com/seshadrilab/CD4-and-CD8-co-receptors-modulate-functional-avidity-of-CD1b-restricted-T-cells (https://doi.org/10.5281/zenodo.5542410). Source data are provided with this paper and on the GitHub repository (Source Data). Source data are provided with this paper.

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

## Acknowledgements

The authors thank Krystle Yu, Paula Marsland, Natalie Erjavec, Sarah Roepke, and Martin Prlic for their technical assistance, D. Branch Moody for supplying purified GMM for these studies, and Stephen De Rosa for supplying the Seattle Assay Control samples. This work was supported by the U.S. National Institutes of Health (R01AI12518904 to C.S.) and the Doris Duke Charitable Foundation (Grant No. 2016103 to C.S.). C.A.J. was supported by the Institute of Translational Health Sciences (5TL1TR00231803) and the Molecular Medicine Training Program (T32GM095421).

## Author contributions

C.S. and C.A.J. conceived of the study. C.A.J., Y.X., E.D.L., M.S.A., L.J., D.M.K., and E.H.W. performed or were involved with experiments. C.A.J. analyzed the data and generated the figures. T.J.S. and C.L.D. facilitated access to archived PBMC and clinical data. A.J.M. and M.G. facilitated access to lipid antigens used to generate reagents for this study. C.A.J. and C.S. wrote and edited manuscript with contributions from all authors.

## Competing interests

The authors declare no competing interests.
