## [Peer Review File · Nature Communications]

CD4 and CD8 co-receptors modulate functional avidity of CD1b-restricted T cellsEditorial Note: This manuscript has been previously reviewed at another journal that is not operating a transparent peer review scheme. This document only contains reviewer comments and rebuttal letters for versions considered at *Nature Communications*. Mentions of prior referee reports have been redacted.

REVIEWER COMMENTS

Reviewer #1 (Remarks to the Author):

James et al address the interesting question as to whether also T cells specific for CD1b-restricted bacterial lipids can be divided into functionally distinct subsets defined by their CD4 vs CD8 coreceptor expression, likewise MHC-restricted T cells. They address this issue with an original combination of experiments tools such as staining of T cells with CD1b multimers loaded with the Mtb lipid antigen SGL, to identify and characterize ex vivo the target cells; CD1b-SGL-restricted TCR transduction into Jurkat cell lines or primary CD4+ or CD8+ T cells, and RNA-seq of single primary CD4+ and CD4- CD1b-SGL-specific T cells. The results provide extensive evidence suggesting that the presence of CD4 or CD8 coreceptors increases the surface expression of the CD1b-SGL-restricted TCRs, and that CD4 or CD8 coreceptor expression enhances the functional avidity of SGL-CD1b-specific T cells, revealing an interesting physiological differences with MHC-restricted T cell subsets.

Major comments

The authors have carefully addressed the queries that were raised by reviewers from another Journal on an previous version of the manuscript. I think that the new version of the manuscript, starting from the title, provides a substantially improved analysis of the results obtained by old and new experiments, conveying a more justified, and novel, message on the function of CD4 and CD8 coreceptors on CD1-restrict T cells specific for lipid antigens, which is different from that on MHC-restricted T cells.

Reviewer #2 (Remarks to the Author):

The MS has undergone significant revisions [REDACTED]. The authors have done a very thorough job of addressing and responding to the points previously noted. The MS is certainly improved with regard to a range of technical concerns. The results are interesting, and overall constitute a substantial contribution to the maturing field of CD1-restricted T cell immunity. I note a few minor issues with the revised MS for consideration by the authors:

1. New text added in lines 60-63, "Most T cells recognize peptide antigens in the context of MHC class I or class II, which express CD8 and CD4 co-receptors, respectively." The sentence is not conveying the intended meaning accurately. It needs to be re-written.
2. Probably should not have abbreviations like "M.tb>" and "M.tb. disease" in the abstract.
3. Line 304, "...compared to -2% lysis". A negative lysis does not make sense conceptually. This is probably no or minimal lysis plus experimental error. Would be better to say no significant lysis was detected.
4. New text in Discussion lines 343 – 352, not sure their data are "in contrast" with earlier report of Sieling et al., or "consistent with" earlier reports on iNKT cells. The mechanism they are proposing does not seem to overlap with what was proposed and studied in these earlier reports. Those papers were attempting to demonstrate a true co-receptor function for CD4 on CD1b restricted T cells (Sieling) or iNKT cells (Thedreuz). Other than the paper from Thedreuz, there seems to be very little data supporting this direct interaction model with strong analogy to CD4-MHCII and CD8-MHCI direct binding. The jury is still out on whether interactions of this sort exist for CD1 (notwithstanding the strong case put forward in 2007 by the Thedreuz paper, which needs confirmation). They may instead want to emphasize that they are really looking at this issue in a different way than the previous studies, and their findings don't address the related issue of possible direct binding of CD4 or CD8 to CD1 molecules.

Reviewer #3 (Remarks to the Author):

The authors thoroughly responded to my comments [REDACTED]. I am quite familiar with Seurat clustering and cluster 0 always contains cells with upregulated genes (perhaps different settings are being used here than is typically used). Also, the clustering looks odd, cluster 1 (green) has cells that are closer to cluster 2 (blue) than to the main blue cluster.

Reviewer #4 (Remarks to the Author):

Comments.

1. Authors identified clusters using either UMAP or Seurat. How robust are these clusters? Some analysis must be performed to ensure that it is not possible to cluster the data differently using either different parameters in these algorithms or if using a different algorithm. In particular, it appears that the method used by the authors uses nearest neighbor clustering. This method may generate biases. These could be useful references to look up: PMID: 22759431, 31874625.
2. Some of the presented analyses of differences between expression of different markers do not properly show paired data, e.g., data in Fig. 1C-E (and similar data later) are per individual, thus, need to be connected by lines with the stat. tests being done appropriately.

Reviewer #1 (Remarks to the Author):

James et al address the interesting question as to whether also T cells specific for CD1b-restricted bacterial lipids can be divided into functionally distinct subsets defined by their CD4 vs CD8 coreceptor expression, likewise MHC-restricted T cells. They address this issue with an original combination of experiments tools such as staining of T cells with CD1b multimers loaded with the Mtb lipid antigen SGL, to identify and characterize ex vivo the target cells; CD1b-SGL-restricted TCR transduction into Jurkat cell lines or primary CD4+ or CD8+ T cells, and RNA-seq of single primary CD4+ and CD4- CD1b-SGL-specific T cells. The results provide extensive evidence suggesting that the presence of CD4 or CD8 coreceptors increases the surface expression of the CD1b-SGL-restricted TCRs, and that CD4 or CD8 coreceptor expression enhances the functional avidity of SGL-CD1b-specific T cells, revealing an interesting physiological differences with MHC-restricted T cell subsets.

We thank the reviewer for this concise and positive summary of our experiments and conclusions.

Major comments

The authors have carefully addressed the queries that were raised by reviewers from another Journal on an previous version of the manuscript. I think that the new version of the manuscript, starting from the title, provides a substantially improved analysis of the results obtained by old and new experiments, conveying a more justified, and novel, message on the function of CD4 and CD8 coreceptors on CD1-restrict T cells specific for lipid antigens, which is different from that on MHC-restricted T cells.

We thank the reviewer for these positive comments on the impact of the results described in our revised manuscript.

Reviewer #2 (Remarks to the Author):

The MS has undergone significant revisions [REDACTED]. The authors have done a very thorough job of addressing and responding to the points previously noted. The MS is certainly improved with regard to a range of technical concerns. The results are interesting, and overall constitute a substantial contribution to the maturing field of CD1-restricted T cell immunity. I note a few minor issues with the revised MS for consideration by the authors:

We appreciate this reviewer's positive comments on our revised manuscript, in particular the recognition that our contributions represent a conceptual advance for the field.

1. New text added in lines 60-63, "Most T cells recognize peptide antigens in the context of MHC class I or class II, which express CD8 and CD4 co-receptors, respectively." The sentence is not conveying the intended meaning accurately. It needs to be re-written.

We apologize for the confusion and agree that grammatically this sentence does not accurately summarize the body of literature. We have revised this sentence as follows:

Lines 61-64: Canonically, T cells recognize peptide antigens in the context of major histocompatibility complex class I (MHC-I) or class II (MHC-II). Recognition of peptides presented by MHC-I and MHC-II is augmented by expression of CD8 or CD4 co-receptors on the surface of the T cell, respectively.

2. Probably should not have abbreviations like "M.tb>" and "M.tb. disease" in the abstract.

We thank the reviewer for this clarifying suggestion and have replaced "M.tb disease" with "tuberculosis" in the abstract.

Lines 50-52: Finally, we used single-cell sequencing to define the TCR repertoire and ex vivo functional profiles of SGL-specific T cells from individuals with *active tuberculosis*.

Lines 56-57: Thus, expression of CD4 and CD8 co-receptor modulates TCR avidity for lipid antigen, leading to functional diversity and differences in in vivo proliferation during *tuberculosis disease*.

3. Line 304, "...compared to -2% lysis". A negative lysis does not make sense conceptually. This is probably no or minimal lysis plus experimental error. Would be better to say no significant lysis was detected.

We agree with this comment that a qualitative statement here would be conceptually simpler when describing these results. We have revised the text as follows:

Lines 298-300: A01 also secreted 10-fold more granzyme B than A05 and specifically lysed 67% of target cells in the presence of antigen, compared to *no significant lysis* in the absence of antigen ($p = 0.03$) (Figures 5B, 5C, and 5D).

4. New text in Discussion lines 343 – 352, not sure their data are "in contrast" with earlier report of Sieling et al., or "consistent with" earlier reports on iNKT cells. The mechanism they are proposing does not seem to overlap with what was proposed and studied in these earlier reports. Those papers were attempting to demonstrate a true co-receptor function for CD4 on CD1b restricted T cells (Sieling) or iNKT cells (Theirez). Other than the paper from Theirez, there seems to be very little data supporting this direct interaction model with strong analogy to CD4-MHCII and CD8-MHCI direct binding. The jury is still out on whether interactions of this sort exist for CD1 (notwithstanding the strong case put forward in 2007 by the Theirez paper, which needs confirmation). They may instead want to emphasize that they are really looking at this issue in a different way than the previous studies, and their findings don't address the related issue of possible direct binding of CD4 or CD8 to CD1 molecules.

We thank this reviewer for these comments on how we contextualized our findings with previous studies and agree that our manuscript does not investigate a direct interaction model for co-receptor-enhanced recognition. We have removed the language directly comparing the findings of this study to previous studies, as the reviewer correctly points out that the differences in methodology between the manuscripts makes mechanistic comparisons challenging. In fact, our methods allowed us to discover a novel mechanism by which co-receptors might be acting to enhance lipid antigen recognition, and this mechanism was not discussed by previous studies.

We included this section to provide context for how little is currently known about the role of co-receptors and lipid antigen recognition, and to highlight that the methods used can significantly affect the

finding, as evidenced by the discrepancy between the findings of Thedrez et al. and Sieling et al. We have modified the text to emphasize these points.

Lines 339-348: ~~Our data contrast with~~ A previous report *found* that the CD4 co-receptor is not involved in lipid antigen recognition by CD1b-restricted T cells (Sieling et al., 2000). This study used blocking antibodies targeting the interaction between CD4 and MHC-II, which may not have reliably predicted the effect on CD1b. Further, the use of co-receptor blocking antibodies can have pleiotropic effects, including paradoxical activation, of T cells (Campanelli et al., 2002). *To avoid this limitation*, we adopted a molecular approach to this question, and our results uncover a role for TCR co-receptors in lipid-specific T cell activation. Previous reports have shown that the CD4 co-receptor enhances iNKT cell activation, *which the authors posit was through direct binding of CD4 to CD1d* (Chen et al., 2007; Thedrez et al., 2007). Our data extend these studies by revealing a potential molecular mechanism in the form of stabilized TCR expression at the surface of CD1b-restricted T cells, which would not have emerged from experiments using blocking antibodies.

Reviewer #3 (Remarks to the Author):

The authors thoroughly responded to my comments. [REDACTED].

I am quite familiar with Seurat clustering and cluster 0 always contains cells with upregulated genes (perhaps different settings are being used here than is typically used). Also, the clustering looks odd, cluster 1 (green) has cells that are closer to cluster 2 (blue) than to the main blue cluster.

We thank this reviewer for the positive comments and apologize for the confusion [REDACTED].

We modeled our analysis pipeline directly from Seurat (Version 3.0) documentation, and selected parameters for resolution and dimensions using the guidelines from the documentation and feature selection suggestions. There is no default setting discussed in the documentation or apparent in the source code that fixes cluster 0 to contain features that are positively enriched (<https://github.com/satijalab/seurat/>) (Stuart, Butler, et al., Cell 2019). In fact, this is one benefit of using unsupervised clustering methods that allow us to detect meaningful differences in gene expression, while remaining agnostic to the gene and direction of change. Perhaps, the focused nature of the method we utilized here where we analyze fewer than 30 genes has allowed us to identify a “negative” cluster where none of the genes assayed were enriched. This is in contrast to RNA sequencing data where the sheer number of genes in these datasets may favor scenarios where upregulated genes are present in each cluster. The final code we used for this analysis is publicly available and has been externally reviewed for accuracy and reproducibility (<https://github.com/seshadri/CD1b-restrictedSingleCellAnalysis>).

We also appreciate this reviewer's in-depth analysis of our UMAP plot and clustering, noting that some cells seem to lie outside of where one could visually define the clusters. This “imperfect clustering” may be a product of low variance between cells, as we used binary expression data on a focused set of features, which is in contrast to single cell RNA sequencing data which is much larger and can be truly quantitative. In addition, when calculating the k-nearest neighbors (KNN) network to assign clusters to cells, Seurat uses a method based on principal components rather than UMAP. Therefore, the clusters may not perfectly match when visualized in UMAP space. However, we would expect the relative relationships to remain intact. To confirm this, we visualized the data by plotting the clusters using principal components, t-SNE, or UMAP and found that the separation of three clusters is preserved across all three methods (Rebuttal Figure 1). In addition, we have performed an additional analysis using hierarchical clustering to validate our clustering approach in our response to Reviewer #4, Comment #1.

Rebuttal Figure 1. Dimensionality reduction of single cell transcriptional profiling data. Spatial relationships between Seurat-defined clusters are preserved when visualizing data using PCA (left), t-SNE (center), and UMAP (right).

Reviewer #4 (Remarks to the Author):

Comments.

1. Authors identified clusters using either UMAP or Seurat. How robust are these clusters? Some analysis must be performed to ensure that it is not possible to cluster the data differently using either different parameters in these algorithms or if using a different algorithm. In particular, it appears that the method used by the authors uses nearest neighbor clustering. This method may generate biases. These could be useful references to look up: PMID: 22759431, 31874625.

We thank this reviewer for their suggestion to use a different algorithm to validate that CD4+ and CD8+ glycolipid-specific T cells cluster separately. We have included an additional analysis in Revised Supplemental Figure 9 where we clustered the cells using hierarchical clustering (complete linkage method) and visualized using a heatmap (Revised Supplemental Figure 9). Of note, CD4+ and CD8+ T cells cluster on the bottom and top of the heatmap, respectively, highlighting the reproducibility of our finding through a complementary clustering algorithm (Revised Supplemental Figure 9).

Line 274-279: We then performed a supervised analysis focusing on cells in which either CD4 or CD8 transcript was detected (105 of 272 total cells) (Supplemental Figure 7, *Supplemental Figure 9*). We identified three distinct clusters using Uniform Manifold Approximation and Projection (UMAP) (Figure 4A) (McInnes et al., 2018). T cells expressing the CD4 and CD8 co-receptor were spatially separated, suggesting that they express unique transcriptional profiles (Figure 4B, *Supplemental Figure 9*).

Lines 1208-1214: (A) Heatmap summarizes expression of genes assayed in SGL-CD1b and GMM-CD1b tetramer-positive cells ($n = 105$). Each row represents one cell. Red indicates that the gene was detected, and yellow indicates that the gene was not detected. A minimum read count of 5 reads per gene was used to define expression. Dendrogram summarizes similarity between cells, where the distance is proportional to the level of dissimilarity. This relationship was computed using complete linkage hierarchical clustering. Dendrogram tips are colored by co-receptor expression pattern (CD4: green, CD8: grey, Both: black).

2. Some of the presented analyses of differences between expression of different markers do not properly show paired data, e.g., data in Fig. 1C-E (and similar data later) are per individual, thus, need to be connected by lines with the stat. tests being done appropriately.

We thank this reviewer for their careful attention to data visualization and choice of statistical test. We agree that visualizing the data with connected lines to emphasize the paired nature of the data adds clarity to our experimental design. We have revisualized our data to address this and updated the Statistical methods section and Figure legend to describe the analyses performed for this visualization, which were the Wilcoxon signed-rank test, which is the appropriate nonparametric test for two-sample paired data, and the Friedman test, which is the appropriate nonparametric test for two-way repeated measures data (Revised Figure 1).

Lines 140-145: SGL-CD1b specific T cells exhibited an 18-fold enrichment of CD4 and CD8 double-positive (DP) cells ($p = 0.003$) and a 1.5-fold enrichment of CD4 T cells ($p = 0.003$) relative to total CD3⁺ T cells. This was accompanied by a 5.6-fold reduction of CD8 cells ($p = 0.003$) and a 2.8-fold reduction of CD4 and CD8 double negative (DN) cells ($p = 0.004$) (Figure 1C). Further, SGL-CD1b specific T cells expressing any combination of co-receptors exhibit consistently higher tetramer MFI when compared to DN SGL-CD1b specific T cells ($p < 0.0001$) (Figure 1D).

Lines 147-151: However, we did detect a 4% lower CD3 ϵ MFI among DN T cells when compared to the other groups ($p = 0.05$) (Figure 1E). For additional context, we quantified the level of CD3 ϵ expression in all DP, CD4, CD8, and DN T cells from these donors and found that CD4 and DN T cells express higher levels of CD3 than CD8 T cells, which is consistent with published literature ($p < 0.0001$ and $p = 0.004$, respectively, Supplemental Figure 1) (El Hentati et al., 2010).

Lines 413-423: Each dot represents the percent of one individual, *and solid lines connect each individual*. The percent of cells in each group was compared to that present in total CD3⁺ T cells (grey). (*Wilcoxon signed-rank test* with Bonferroni Correction, $** = 0.0001 < p < 0.001$, $n = 15$). (D) Boxplots depict the median and interquartile range of mean fluorescence intensity (MFI) of SGL-CD1b tetramer-positive cells in each co-receptor group. Each dot represents the MFI of one sample, *and solid lines connect each individual*. MFI was compared between groups (*Friedman test* with post-hoc Dunn test, $*** = p < 0.0001$, $** = 0.0001 < p < 0.001$, $n = 15$). (E) Boxplots depict the median and interquartile range of mean fluorescence intensity (MFI) of CD3 among tetramer-positive cells in each co-receptor group. Each dot represents the MFI of one sample, *and solid lines connect each individual*. MFI was compared between groups (*Friedman test* with post-hoc Dunn test, $* = 0.001 < p < 0.05$, $n = 15$).

Lines 840-843: A Wilcoxon Signed Rank test *or a Friedman test* was used when the continuous variables were not independent for *two groups and three or more groups, respectively*. Post-hoc Dunn tests were performed after ANOVA, Kruskal-Wallis, *and Friedman tests* to determine which group(s) were different.

Lines 1112-1116: (D) Boxplots depict the median and interquartile range of CD3 MFI of all T cells within each co-receptor group (double positive (DP), CD4, CD8, and double negative (DN)). Each dot represents the percent of one sample, *and solid lines connect each individual*. The percent of cells in each group was compared to that present in total CD3⁺ T cells (grey). (*Friedman test* with Dunn post-test, $*** = p < 0.0001$, $* = 0.001 < p < 0.05$, $n = 15$).

REVIEWER COMMENTS

Reviewer #3 (Remarks to the Author):

The problem has been that the methods, results, figure legends were not clear to me until now. The authors state in the response "Perhaps, the focused nature of the method we utilized here where we analyze fewer than 30 genes has allowed us to identify a "negative" cluster where none of the genes assayed were enriched." The data and results section look very much like a single cell RNA sequencing approach to measure the whole transcriptome, but I recognize that less than 30 genes were examined as shown in supplementary tables 5 and 6, which paradoxically come after the data in supplementary table 4 as the methods are at the back of the paper. I think this could be easily clarified if the exact number of genes studied is clear both in the results section and methods.

Reviewer #4 (Remarks to the Author):

I appreciate the authors to use another method to cluster their data. But I am still left unclear on several points about clustering.

1) why is the number of clusters what it is? (i.e., 7 or 3 - how is that statistically determined? Why not 6 and 4?). This needs to be very well justified. 2) There is not statistical comparison of clustering obtained by 2 alternative methods. There is no rigorous analysis that these two methods give statistically similar result. 3) Are the clusters robust? This can be investigated by using resampling?

The authors updated the figures to show paired data in Figure 1. But this was only done for Figure 1 while paired data are presented in several other figures. Please correct ALL figures that contain paired data (i.e., I don't think I need to search myself which data are paired and which are not, authors need to determine that and make the appropriate plots indicating this).

Reviewer #3 (Remarks to the Author):

The problem has been that the methods, results, figure legends were not clear to me until now. The authors state in the response "Perhaps, the focused nature of the method we utilized here where we analyze fewer than 30 genes has allowed us to identify a "negative" cluster where none of the genes assayed were enriched." The data and results section look very much like a single cell RNA sequencing approach to measure the whole transcriptome, but I recognize that less than 30 genes were examined as shown in supplementary tables 5 and 6, which paradoxically come after the data in supplementary table 4 as the methods are at the back of the paper. I think this could be easily clarified if the exact number of genes studied is clear both in the results section and methods.

We thank the reviewer for these suggestions to improve the clarity of our experimental workflow. We have expanded our discussion of the single-cell methods in the results section to highlight the number of genes and workflow in the main text.

Lines 267-271: To examine whether the expression of CD4 or CD8 co-receptors was associated with different functional profiles, we used *targeted amplification of 23 genes* to examine expression of these transcripts by GMM- and SGL-specific T cells (n = 223) and tetramer-negative T cells (n = 43) *in two individuals with active tuberculosis*. We also included T cell lines with known functional profiles as positive controls (n = 6) (Han et al., 2014) (Supplemental Tables 4 and 5).

Reviewer #4 (Remarks to the Author):

I appreciate the authors to use another method to cluster their data. But I am still left unclear on several points about clustering.

1) why is the number of clusters what it is? (i.e., 7 or 3 - how is that statistically determined? Why not 6 and 4?). This needs to be very well justified. 2) There is not statistical comparison of clustering obtained by 2 alternative methods. There is no rigorous analysis that these two methods give statistically similar result. 3) Are the clusters robust? This can be investigated by using resampling?

We would like to thank this reviewer for the suggestions on how to improve analysis of our single-cell gene expression data, specifically the use of Seurat, which was originally developed for clustering high-dimensional RNA-Seq data. We agree that additional analysis would be necessary to verify the robustness of the clusters, particularly in light of the small sample size and relatively low dimensionality (23 genes) compared to RNA-Seq. We explored several approaches to doing this within Seurat, including permutation and resampling, as suggested by the Reviewer. Ultimately, we decided that readers would be too confused by our non-standard approach to use Seurat for this analysis and opted instead for a simpler, more transparent, and ultimately more conservative approach.

In short, we performed a univariate analysis by comparing the expression of each gene between CD4 and CD8 CD1b-restricted T cells using a Fisher's exact test and then correcting for multiple comparisons using the Benjamini-Hochberg method. We found that CD8 T cells expressed higher levels of TBET and PRF-1 compared to CD4 T cells (adjusted p-value 0.02 and 0.04, respectively) and found higher levels of BCL6 in CD4 T cells compared to CD8 T cells (adjusted p-value 0.02). These results confirm the findings from our cluster-based analysis by showing that CD8 mycolipid-specific T cells are enriched for cytotoxic phenotypes, while CD4 T cells exhibit more functional heterogeneity. The results also strengthen our central claim that differences in TCR avidity between CD4 and CD8 mycolipid-specific T cells is accompanied by differences in phenotype. In revising the manuscript to reflect the new approach and results, we have also enhanced our discussion of all the phenotypes discovered by our approach, irrespective of co-receptor status, as these data are among the first to

profile mycobacterial lipid-specific T cells using a single-cell approach directly ex vivo in the context of active TB disease.

We have modified the Abstract, Methods, Results, and Discussion and included a revised Figure 4 and new Supplementary Table 4.

Lines 52-56: *We found that CD8+ T cells specific for SGL express canonical markers associated with cytotoxic T lymphocytes (PRF-1 and T-BET) compared to CD4+ T cells. We also detected Ki-67 expression among SGL-specific T cells, suggesting that they were actively proliferating at the time of sample collection. Thus, expression of CD4 and CD8 co-receptor modulates TCR avidity for lipid antigen, leading to in vivo functional diversity during tuberculosis disease.*

Lines 271-289: *We performed a supervised analysis focusing on cells in which either CD4 or CD8 transcript was detected (104 of 272 total cells). We first utilized hierarchical clustering to visualize the overall diversity as well as the relationships between CD4+ and CD8+ CD1b-restricted T cells. T cells expressing the CD4 and CD8 co-receptor were spatially separated, suggesting that they express unique transcriptional profiles (Figure 4A). Of note, we detected expression of all master transcription factors that regulate TH1 (TBET), TH2 (GATA3), TH17 (RORC), and Treg (FOXP3) functional profiles, indicating previously undescribed functional diversity among CD1b-restricted T cells (Figure 4A, 4B). Of note, MKI67 was expressed in 14.2% of CD1b-restricted T cells that were included in our final analysis and 0% of tetramer-negative T cells, which suggests that CD1b-restricted T cells were proliferating at the time of sample collection and that Ki-67 expression is not generally expressed in peripheral blood T cells in individuals with active TB ($p = 0.12$) (Figure 4A).*

We next sought to identify the differentially expressed genes (DEGs) driving the differences between CD4 and CD8 CD1b-restricted T cells (Figure 4C-F). We found that TBET and PRF1 were enriched in CD8 T cells compared to CD4 T cells, and noted a trend toward enrichment of EOMES among CD8 T cells, suggesting cytotoxic phenotypes among CD8 mycolipid-restricted T cells (Figure 4C-E, Supplemental Table 4) ($p = 0.02, 0.04, 0.08$, respectively). We also detected enrichment of the TFH transcription factor BCL6 among CD4 T cells, suggesting functional heterogeneity among CD4 CD1b-restricted T cells (Figure 4F, Supplemental Table 4) ($p = 0.02$).

Lines 354-359: *Our data reveal previously unappreciated functional diversity of CD1b-restricted T cells as we detected expression of TBET, GATA3, RORC, and FOXP3 among circulating CD1b-restricted T cells ex vivo. Mycobacterial glycolipid-specific T cells that express the CD8 co-receptor are broadly cytotoxic T cells. This is consistent with the original description of an SGL-specific T cell clone, which produced IFN- γ in the presence of M.tb infected cells and reduced the bacterial burden of M.tb in culture (Gilleron et al., 2004). Conversely, T cells that express the CD4 co-receptor are more diverse.*

Lines 370-376: *Together, our data highlight the importance of considering co-receptor expression by CD1b-restricted T cells when evaluating lipid-containing vaccines in clinical and pre-clinical models as CD4 T cells may be more likely to be activated in the context of infection, possibly as a result of enhanced functional avidity. Lipid-based vaccines have shown promise in pre-clinical animal models (Shang et al., 2018). CD4 or CD8 CD1b-restricted T cells could be differentially targeted by the adjuvant or delivery platform when 'tuning' the chemical structure of the antigen to modify TCR avidity and immunogenicity of these novel vaccine strategies.*

Lines 437-450: *Figure 4. Gene expression differences between ex vivo CD4 and CD8 CD1b-restricted T cells. (A) Heatmap summarizes expression of genes assayed in SGL-CD1b and GMM-CD1b tetramer-positive cells ($n = 104$). Each row represents one cell. Red indicates that the gene was detected, and blue indicates that the*

gene was not detected. A minimum read count of 5 reads per gene was used to define expression. Dendrogram summarizes similarity between cells, where the distance is proportional to the level of dissimilarity. This relationship was computed with complete linkage hierarchical clustering using binary distance metric. Dendrogram tips are colored by co-receptor expression pattern (CD4: black, CD8: grey). (B) Heatmap summarizes expression of transcription factors assayed in SGL-CD1b and GMM-CD1b tetramer-positive cells and tetramer-negative T cells ($n = 104$). Each row represents one cell. Red indicates that the gene was detected, and blue indicates that the gene was not detected. A minimum read count of 5 reads per gene was used to define expression. (C-F) Bar charts summarize the percentage of CD4 (black) and CD8 (white) T cells expressing TBET (C), PRF1 (D), EOMES (E), and BCL6 (F) ($n = 104$) (Fisher's exact test, Benjamini-Hochberg corrected). Summary of statistical tests is included in Supplemental Table 4.

Lines 790-797: Data were then input into the R programming environment and further processed and formatted. The resulting matrix was then analyzed using hierarchical clustering to visualize relationships between functional profiles expressed by T cells where CD4 or CD8 were detected ($n = 104$ of 272 total sorted cells). For T cells where both CD4 and CD8 were detected, these were coded as "CD4", as most exhibited a bias toward CD4 expression and biologically CD4 T cells that also express CD8a are likely to be activated CD4 T cells (Kenny et al., 2004). Fisher's Exact Tests were used to determine which genes were significantly enriched among CD4 or CD8 T cells and p -values were adjusted for multiple comparisons using the Benjamini-Hochberg method

Lines 1049-1052: Supplemental Table 4. Differentially expressed genes among CD4 and CD8 CD1b-restricted T cells. P -values for each gene are summarized here P -values were calculated for each gene (Feature) between CD4 and CD8 T cells using following a Fisher's Exact Test (p -value) and Benjamini-Hochberg correction (adjusted p -value) ($n = 23$ features).

The authors updated the figures to show paired data in Figure 1. But this was only done for Figure 1 while paired data are presented in several other figures. Please correct ALL figures that contain paired data (i.e., I don't think I need to search myself which data are paired and which are not, authors need to determine that and make the appropriate plots indicating this).

We thank the author for these suggestions for our data visualization. We reviewed every figure panel and considered whether a change in visualization was appropriate based on the nature of the data and whether such a change would clarify the interpretation. In short, we did not feel that any additional modifications were necessary beyond those already made in Figure 1 and Supplemental Figure 1.

Figure 1A-C: Not applicable, representative gating strategy

Figure 1D-E: These boxplots summarize the surface phenotypes of T cell subsets identified from a group of donors. This is paired data and the visualization was modified in the previous iteration of this manuscript.

Figure 2A: Not applicable, representative gating strategy

Figure 2B: This figure compares distributions of SGL-CD1b tetramer MFI among transduced T cells from one individual. This is not amenable to this reviewer's suggestion as the data are not summarized by individual.

Figure 2C: This histogram represents two independent transduced T cell clones derived from the same donor.

Figure 2D: These boxplots summarize SGL-CD1b tetramer MFI and CD3 expression of 16 independent transduced T cell clones derived from the same donor.

Figure 2E: This graph summarizes activation data from two independent transduced T cell clones derived from the same donor.

Figure 3A: Not applicable, representative gating strategy

Figure 3B: These data are derived from two independent Jurkat lines transduced with the same TCR.

Figure 3C: Left: Not applicable, representative gating strategy. Right/Center: These boxplots summarize SGL-CD1b tetramer MFI and CD3 expression of Jurkats that are derived from the same line and same experimental well. These are paired data. We have included boxplots below that contain the visualization suggestion of connecting each experimental well with a line (Figure 1). Due to the low variance between the replicates, we do not think this change enhances the interpretation of these data and decided not to include this suggestion in the manuscript.

Figure R1. Modified plots for Figure 3C. Reproduced plots from Figure 3C with solid black lines connecting points from each well. Due to low variance between wells, this addition does not add clarity to these plots and thus was not included in the final manuscript.

Figure 4A: Not applicable, heatmap

Figure 4B: Not applicable, heatmap

Figure 4C-F: Data are summarized from binary expression data of cells and not summarized by individual. Adding lines to connect individuals is not appropriate for this type of visualization.

Figure 5A: This barplot is representative of cytokine expression by two independent T cell lines.

Figure 5B: This graph is representative of cytokine expression by two independent T cell lines.

Figure 5C: Not applicable, representative gating strategy.

Figure 5D: This boxplot is representative of cytotoxic activity by two independent T cell lines.

Supplemental Figure 1A: Not applicable, representative gating strategy.

Supplemental Figure 1B: Not applicable, representative gating strategy.

Supplemental Figure 1C: This boxplot is summarized by cohort group, and individuals are not represented in multiple cohorts.

Supplemental Figure 1D: This boxplot summarizes CD3 MFI by mycolipid-specific T cells. As the individuals are represented in each group, this represents paired data. This was edited in the previous version of the manuscript.

Supplemental Figure 1E-H: Not applicable, representative gating strategy.

Supplemental Figure 2A-B: Both bar plot and box plot summarize SGL-CD1b tetramer staining of two independent T cell lines.

Supplemental Figure 3: Not applicable, representative gating strategy.

Supplemental Figure 4: This figure compares distributions of SGL-CD1b tetramer MFI among one individual. This is not amenable to this reviewer's suggestion as there are not multiple individuals represented in these data.

Supplemental Figure 5: Data are summarized from expression data of cells and not summarized by individual. Adding lines to connect individuals is not appropriate for this type of visualization.

Supplemental Figure 6: Data are summarized from expression data of cells and not summarized by individual. Adding lines to connect individuals is not appropriate for this type of visualization.

Supplemental Figure 7: Not applicable, representative gating strategy.

REVIEWER COMMENTS

Reviewer #3 (Remarks to the Author):

The authors state in response to reviewer 4:

"We would like to thank this reviewer for the suggestions on how to improve analysis of our single-cell gene expression data, specifically the use of Seurat, which was originally developed for clustering high dimensional RNA-Seq data. We agree that additional analysis would be necessary to verify the robustness of the clusters, particularly in light of the small sample size and relatively low dimensionality (23 genes) compared to RNA-Seq. We explored several approaches to doing this within Seurat, including permutation and resampling, as suggested by the Reviewer. Ultimately, we decided that readers would be too confused by our non-standard approach to use Seurat for this analysis and opted instead for a simpler, more transparent, and ultimately more conservative approach."

The paper originally and still reports single cell profiling, but it is now clear that this was done with 23 genes. The readers of nature communications would expect that single cell profiling reports scRNA-seq data. The original UMAP type plots have been replaced with heat maps which are not impressive. Yet they conclude that the cells are functionally diverse, based on 23 genes that they picked. They cannot use Seurat to analyze their data as they say it would be unconventional. Well their approach is unconventional for 2021.

I would never advocate rejecting a paper at this stage. Nature Medicine did this to me over an isotype control antibody and I still am unhappy about that. Perhaps the answer is that they could soften their conclusion about functional diversity since they only measured 23 genes. Something like- the data show functional diversity in the context of measuring 23 genes. I don't think this part of the paper is key for acceptance.

Reviewer #4 (Remarks to the Author):

All changes are good, thank you for taking them seriously.

REVIEWERS' COMMENTS

Reviewer #3 (Remarks to the Author):

The authors state in response to reviewer 4:

"We would like to thank this reviewer for the suggestions on how to improve analysis of our single-cell gene expression data, specifically the use of Seurat, which was originally developed for clustering high dimensional RNA-Seq data. We agree that additional analysis would be necessary to verify the robustness of the clusters, particularly in light of the small sample size and relatively low dimensionality (23 genes) compared to RNA-Seq. We explored several approaches to doing this within Seurat, including permutation and resampling, as suggested by the Reviewer. Ultimately, we decided that readers would be too confused by our non-standard approach to use Seurat for this analysis and opted instead for a simpler, more transparent, and ultimately more conservative approach."

The paper originally and still reports single cell profiling, but it is now clear that this was done with 23 genes. The readers of nature communications would expect that single cell profiling reports scRNA-seq data.

We agree that this nomenclature has caused significant confusion and we have replaced all references to single cell profiling with "targeted transcriptional profiling."

Lines 50-52: *Targeted transcriptional profiling of mycolipid-specific T-cells from individuals with active tuberculosis revealed canonical markers associated with cytotoxicity (PRF-1 and T-BET) among CD8+ compared to CD4+ T-cells.*

Lines 101-103: *Targeted transcriptional profiling studies of SGL-specific T cells revealed distinct TCR repertoires and functional programs that could be classified on the basis of CD4 and CD8 expression.*

Lines 285-287: Taken together, these data support the conclusions of our *targeted transcriptional profiling* experiments and show that SGL-specific T cells expressing CD4 generally express cytokines that align with a T-helper phenotype, while those expressing CD8 have a cytotoxic effector phenotype.

The original UMAP type plots have been replaced with heat maps which are not impressive. Yet they conclude that the cells are functionally diverse, based on 23 genes that they picked. They cannot use Seurat to analyze their data as they say it would be unconventional. Well their approach is unconventional for 2021.

We appreciate the feedback that the heatmaps as displayed are not visually impressive. We have re-analyzed our heatmap and included clustering on the x-axis in addition to the existing clustering on the y-axis, per the reviewer's suggestion, and believe this enhances visualization of the expression differences of the genes analyzed between CD4 and CD8 T cells. We have also removed Figure 4B from the manuscript, as this reviewer correctly pointed out that CD4 and CD8 are not described here, and as such this figure does not add significant value to the manuscript.

Lines 258-261: Of note, we detected expression of all master transcription factors that regulate TH1 (TBET), TH2 (GATA3), TH17 (RORC), and Treg (FOXP3) functional profiles, indicating previously undescribed functional diversity among CD1b-restricted T cells (Figure 4A, 4B).

Lines 438-445: Dendrograms on x and y axis summarize similarity between cells (y) or genes (x), where the distance is proportional to the level of dissimilarity. This relationship was computed with complete linkage hierarchical clustering using binary distance metric. Y-axis dendrogram tips are colored by co-receptor

expression pattern (CD4: black, CD8: grey), and *CD4* and *CD8* genes on the X-axis have been bolded for emphasis. ~~(B) Heatmap summarizes expression of transcription factors assayed in SGL-CD1b and GMM-CD1b tetramer-positive cells and tetramer-negative T cells (n = 104). Each row represents one cell. Red indicates that the gene was detected, and blue indicates that the gene was not detected. A minimum read count of 5 reads per gene was used to define expression.~~

I would never advocate rejecting a paper at this stage. Nature Medicine did this to me over an isotype control antibody and I still am unhappy about that. Perhaps the answer is that they could soften their conclusion about functional diversity since they only measured 23 genes. Something like- the data show functional diversity in the context of measuring 23 genes. I don't think this part of the paper is key for acceptance.

We thank this reviewer for explicitly indicating that this is not grounds for rejection. We have softened our conclusion statement in the Discussion to specifically indicate that we demonstrated “functional diversity” by assaying a targeted panel of genes, and adding that additional diversity could be potentially uncovered by performing scRNA-Seq.

Lines 355-359: Our data reveal previously unappreciated functional diversity of CD1b-restricted T cells as we detected expression of TBET, GATA3, RORC, and FOXP3 among circulating CD1b-restricted T cells ex vivo. Mycobacterial glycolipid-specific T cells that express the CD8 co-receptor are broadly cytotoxic T cells. *Evaluation of the whole transcriptome may reveal additional functional diversity we did not detect due to our targeted approach.*

Reviewer #4 (Remarks to the Author):

All changes are good, thank you for taking them seriously.

We thank this reviewer for their input during the review process and are pleased that we have thoroughly addressed their concerns in full.